



# Stable isotope signatures of Holocene syngenetic permafrost trace seabird presence in the Thule District (NW Greenland)

Sebastian Wetterich[1], Thomas A. Davidson[2], Anatoly Bobrov[3], Thomas Opel[1], Torben Windirsch[1], Kasper L. Johansen[4], Ivan González-Bergonzoni[5], Anders Mosbech[4], Erik Jeppesen[2]

5 [1] Alfred Wegener Institute, Helmholtz Center for Polar and Marine Research, Potsdam, Germany
[2] Department of Bioscience, Arctic Research Centre, Aarhus University, Silkeborg, Denmark
[3] Department of Soil Science, Moscow State University, Russia
[4] Department of Bioscience, Arctic Research Centre, Aarhus University, Roskilde, Denmark
[5] Departamento del Agua, CENUR Litoral Norte, Universidad de la República, Paysandú, Uruguay

*Correspondence to:* Sebastian Wetterich (sebastian.wetterich@awi.de)

**Abstract.** Holocene permafrost from ice wedge polygons in the vicinity of large seabird breeding colonies in the Thule District, NW Greenland, was drilled to explore the relation between permafrost aggradation and seabird presence. The latter is reliant on the presence of the North Water (NOW) polynya in the northern Baffin Bay. The onset of peat accumulation associated 15 with the arrival of little auks (*Alle alle*) in a breeding colony at Annikitisoq north of Cape York is radiocarbon-dated to 4400 cal yr BP. A thick-billed murre (*Uria lomvia*) colony on Appat (Saunders Ø) in the mouth of the Wolstenholme Fjord started 5650 cal yr BP. Both species provide marine-derived nutrients (MDNs) that fertilise vegetation and promote peat growth. The geochemical signature of organic matter left by the birds is traceable in the frozen Holocene peat. The peat accumulation rates at both sites are highest after the onset and decrease over time and were about two-times faster at the little auk site than at the 20 thick-billed murre site. High accumulation rates induce shorter periods of organic matter (OM) decomposition before it enters the perennially frozen state. This is seen in comparably high C/N and less depleted $\delta^{13}$C, pointing to a lower degree of OM decomposition at the little auk site, while the opposite pattern can be discerned at the thick-billed murre site. Peat accumulation rates correspond to $\delta^{15}$N trends, where decreasing accumulation led to increasing depletion in $\delta^{15}$N as seen in the little-auk related data. In contrast, the more decomposed OM of the thick-billed murre site shows almost stable $\delta^{15}$N. Late Holocene 25 wedge ice fed by cold season precipitation was studied at the little auk site and provides the first such stable-water isotopic records from Greenland with mean $\delta^{18}$O of -18.0±0.8 ‰, mean δD of -136.2±5.7 ‰, mean *d* excess of 7.7±0.7 ‰, and a $\delta^{18}$O-δD slope of 7.27, which is close to those of the modern Thule Meteoric Water Line. The syngenetic ice wedge polygon development is mirrored in testacean records of the little auk site and delineates polygon low-centre, dry-out and polygon-high-centre stages. The syngenetic permafrost formation directly depending on peat growth (controlled by bird activity) falls 30 within the period of Neoglacial cooling and the establishment of the NOW polynya, thus indirectly follows the Holocene climate trends.

Key words: North Water polynya, Baffin Bay, ice wedge, polygon tundra, little auk, thick-billed murre, nitrogen isotopes, carbon isotopes



# 1 Introduction

The Thule District of Northwest Greenland (Figure 1) is notable for its large seabird colonies along the coast, especially the little auk, *Alle alle* (e.g. Boertmann and Mosbech, 1998), and the thick-billed murre, *Uria lomvia* (e.g. Kampp, 1990), both belonging to the auk family (Alcidae). The mainly piscivorous thick-billed murre and the planktivorous little auk both depend
on the resources from the productive North Water (NOW) polynya, which is situated here in northern Baffin Bay (Stirling, 1980; Davidson et al., 2018). The area is especially important for the little auk as >80% of the global breeding population of the species, corresponding to approximately 33 million pairs, is estimated to breed here (Boertmann and Mosbech, 1998; Kampp et al., 2000; Egevang et al., 2003). This has been linked to an exceptional abundance of their preferred zooplankton prey (*Calanus hyperboreus* and *C. glacialis*), which is available in the NOW throughout the little auk breeding season (Møller
et al., 2018).

The seabirds transport large quantities of marine-derived nutrients (MDNs) from their feeding grounds at sea to the terrestrial coastal environments surrounding their breeding sites. By doing so, they act as ecosystem engineers transforming the areas around the colonies, leaving explicit signatures of their presence (González-Bergonzoni et al., 2017). The input of MDNs around large breeding sites facilitates vegetation growth and peat accumulation at high latitudes (e.g. Zwolicki et al., 2013),
completely altering the prevailing ice-free coastal area that is otherwise dominated by bedrock. Hence, areas around bird colonies are hotspots of floral and faunal biodiversity and productivity (Mosbech et al., 2018) and the onset of peat formation and syngenetic permafrost aggradation coincides with the presence of seabird colonies.

High-latitude soils affected by avian MDN input are enriched in limiting plant nutrients such as nitrogen and phosphorus (e.g. Zwolicki et al., 2013), leading to increased diversity, biomass and coverage of vegetation. Soil-inhabiting protist communities
such as testate amoebae (testaceans) show altered species richness and composition if their habitats are influenced by seabird colonies (Mazei et al., 2018; Vincke et al., 2007).

Vegetation fertilised by MDNs and the subsequent peat formation over millennia enables permafrost aggradation. Syngenetic (concurrent with accumulation) freezing of the peat and ice wedge polygon formation shape the topography when ice wedges grow in polygonal patterns owing to thermal contraction (frost) cracking of the frozen ground in winter and filling of the cracks
mostly by snowmelt in spring (Leffingwell, 1915). The melt water entering the frost cracks refreezes immediately due to the sub-zero ground temperatures and forms vertical ice veins. Ice wedges widen by repetition of these processes and grow upward with ongoing organic or mineralic accumulation on top. The stable isotope composition of water from wedge ice serves as a proxy of winter climate conditions since the ice derives mainly from winter precipitation (e.g. Opel et al., 2018).

Previous studies have examined the current relation between the NOW polynya and little auk populations (Mosbech et al.,
2018), including the availability of their preferred zooplankton prey (e.g. Møller et al., 2018). The onset of seabird colonies in the study area was previously explored using lake sediments and frozen peat by Davidson et al. (2018), whose data are partly employed in the present study that focuses on resulting peat growth and permafrost aggradation in connection with seabird breeding sites. By employing for the first time Holocene permafrost and testacean records from the Thule District the study



aims deciphering (1) the relation of the seabird colony presence and syngenetic permafrost aggradation onset over time, and (2) the alternation and preservation of bird presence signals in OM properties of frozen peat.

## 2 Study area

The climate conditions as recorded in Pituffik (Thule Air Base, TAB, 1948-2013, WMO station 4202) display mean annual air temperatures of -10.8 °C with the warmest mean temperatures in July (5.5°C) and the coldest in February (-25.1 °C) (Cappelen [ed.], 2014). The mean annual precipitation amounts to only 127 mm and the wettest month is August with a mean value of 25 mm (1961-1990, DMI, 2018). The low precipitation and the large annual temperature amplitude create continental climate conditions close to those of High Arctic deserts. The ice-free coastal areas belong to the continuous permafrost zone (Brown et al. [eds.], 1997). Permafrost reaches more than 300 m depth (Roethlisberger, 1961) and the zero-amplitude ground temperature of -10°C is at about 7.6 m below the surface (bs) in the Pituffik Valley (Bjella, 2013). Main parts of the regional permafrost formed epigenetically when sediments and rock froze after Late Glacial to Holocene deglaciation and exposition to negative annual air temperatures. The uppermost permafrost of depositional areas, however, formed syngenetically with ongoing sedimentation and shaped the modern topography by periglacial landforms such as ice wedge polygons and other patterned-ground features (Horwarth et al., 2008). The uppermost and seasonally thawed (active) layer reaches 0.1 to 0.7 m thickness.

The sample material was retrieved at two locations, at Annikitisoq (informal name Great Lake, 76.03288°N, 67.61811°W) north of Cape York and on Appat (Saunders Ø, 76.56908°N, 70.04099°W) in the mouth of the Wolstenholme Fjord (Figure 1). Both sites exhibit well-expressed high-centre polygons that reach about 12 m in diameter with trenches up to 2 m deep at Annikitisoq and about 4 to 5 m in diameter with trenches up to 1.5 m deep on Appat (Figure 2). At Annikitisoq, the polygons partly cover gentle slopes below a colony of little auks and the catchments of a proglacial lake and a stream draining the lake (Figure 2). On Appat, elevated polygon centres were found below a cliff with breeding sites of thick-billed murre. According to the Circumpolar Arctic Vegetation Map, both sites are categorised as prostrate/hemiprostrate dwarf-shrub tundra (P2) composed of dwarf shrubs, mosses, rushes, forbs and lichens (CAVM Team, 2003).

## 3 Material and methods

### 3.1 Fieldwork

Fieldwork was undertaken between 21 July and 14 August 2015 in the Thule District, NW Greenland. The centres of high-centre polygons were chosen for coring. The uppermost active layer was excavated using spades and sampled. The active layer depth reached 20 cm at Annikitisoq and 10 cm on Appat. Below the permafrost table, the peat deposits were extracted using a SIPRE corer driven by a two-stroke engine (Hughes and Terasmae, 1963). The diameter of the coring barrel is 48 mm and its





length 1 m. Extensions were used to reach deeper deposits until the corer hit bedrock material at the base of the peat. The drilling at Annikitisoq (core GL3) reached a depth of 320 cm bs and on Appat (core SI1) 195 cm bs (Table 1). After drilling, cryostructures of intrasedimental ice were described following French and Shur (2010), and the cores were kept frozen until subsampling and analyses. At Annikitisoq, wedge ice was exposed in a trench between two high polygon centres at the cored

GL3 polygon. Neither a distinct frost crack nor rejuvenation stages were observed in the ice wedge. Thus, no traces of modern ice wedge growth were seen and the ice represents the buried remains of a degrading ice wedge network. Clearly expressed shoulders confirm the syngenetic formation of the wedge ice. The wedge ice exposure was about 1.7 m wide, about 1 m high above the trench bottom and about 1.2 m below the GL3 polygon surface (Figure 3). Using axe and hammer, the ice was sampled in 35 subsamples across a horizontal transect between 30 and 50 cm above the trench bottom (at 50 to 80 cm bs).

**3.2 Permafrost analyses**

The frozen cores were sectioned at 2-4 cm increments by a band saw for further analytical work and then freeze-dried. The GL3 core totalled 75 samples (including four unfrozen active-layer samples) and the SI1 core 49 samples (including one unfrozen active-layer sample). The gravimetric ice content was measured as the weight difference between fresh and freeze-dried bulk sediment samples and is expressed as ice content in weight percentage (wt%).

The oxygen ($\delta^{18}O$) and hydrogen ($\delta D$) stable isotope composition of wedge ice was measured using a Finnigan MAT Delta-S mass spectrometer, $1\sigma$ better than 0.1‰ for $\delta^{18}O$ and 0.8‰ for $\delta D$ (Meyer et al., 2000). Values are given as per mil (‰) difference from the Vienna Standard Mean Ocean Water (VSMOW) standard. The deuterium excess ($d$) is calculated following Eq. (1) (Dansgaard, 1964): $d = \delta D - 8 * \delta^{18}O$.

**3.3 Radiocarbon dating and organic matter analyses**

In total, 11 samples from the GL3 core and six samples from the SI1 core were radiocarbon-dated at the Aarhus AMS 14C Centre (AARAMS). The dates are published by Davidson et al. (2018) where more details on lab procedures are given. The radiocarbon ages were transformed into calendar years using the IntCal13 calibration curve (Reimer et al., 2013). Age models for the cores were fitted using the R routine BACON, a Bayesian age-depth modelling approach (Blaauw and Christen, 2011). Ages reported and used in the figures are median modelled ages with $1\sigma$ probability range in calibrated years before the present

(cal yr BP).

Total nitrogen (TN) and total organic carbon (TOC) contents were measured with elemental analysers (ElementarVario EL III for TN and ElementarVario MAX C for TOC; analytical accuracy ± 0.1 wt%). The ratio of TOC and TN is referred to as C/N. Stable carbon ($\delta^{13}C$) and nitrogen ($\delta^{15}N$) isotope analysis was undertaken using a Thermo Scientific Delta V Advantage Isotope Ratio MS equipped with a Flash 2000 Organic Elemental Analyser using helium as a carrier gas. Values are given as per mil





(‰) difference from the Vienna Pee Dee Belemnite (VPDB) standard for $\delta^{13}$C and from nitrogen in ambient air (AIR) for $\delta^{15}$N. The accuracy was better than ± 0.15% for $\delta^{13}$C and ± 0.2% for $\delta^{15}$N. Further details are given in Davidson et al. (2018).

### 3.4 Testacean analysis

Samples for testacean analysis were suspended in purified water and wet-sieved through a 500-µm screen. Testacean shells
were identified on glycerin slides under a light microscope and ecologically characterised following Chardez (1965), Beyens and Bobrov (2016), Bobrov et al. (2013). Species of the eurybiontic, moss-inhabiting and hydrophilic ecological groups were distinguished. The eurybiontic species occur in soils and tolerate a wide range of environmental conditions. Moss-inhabiting species were further distinguished in xerophilic or hygrophilic ecological groups indicating dry or wet conditions, respectively. The hydrophilic group comprises species that inhabit open water bodies. Changes in the ratio of ecological groups indicate
changes in environmental conditions, primarily habitat moisture.

### 4 Results

#### 4.1 Annikitisoq - core GL3 and wedge ice GL3 data

The polygon development at Annikitisoq covers the period from 4400 to 540 cal yr BP over the total core length of 320 cm and was divided into three periods as follows: GL3-I (4400 to 3520 cal yr BP, 320 to 129 cm bs), GL3-II (3520 to 2620 cal yr
BP, 129 to 58 cm bs) and GL3-III (2620 to 540 cal yr BP, 58 to 0 cm bs) (Table 2).

Zone GL3-I is composed of reddish light-brown partly stratified peat with non-parallel, wavy lenticular and irregular reticulated cryostructures. Single ice lenses were > 1 mm thick. The zones GL3-II and GL3-III (except for the uppermost 20 cm active layer) comprise frozen brown peat with the same cryostructures as in GL3-I. The ice content is high in the frozen core with a mean value of 80 ± 4 wt%, while the moisture content of the active layer decreases to 39 wt% in the uppermost
sample. The accumulation rate reflects fast polygon growth after establishment of the little auk colony and decreases in a stepwise manner over time (Figure 4). TN and TOC show low variation over the entire record with mean values of 1.0 ± 0.3 wt% and 45.1 ± 1.8 wt%, respectively. The C/N increases in GL3-I bottom-up from 28 up to a maximum value of 92. Within GL3-II, the C/N varies considerably between 39 and 85 and decreases in GL3-III from 62 to 27 with deceasing depth. The stable carbon and nitrogen composition of the GL3 core shows some variations in zone GL3-I with a mean of -23.0 ± 1.3 ‰
in $\delta^{13}$C and a mean of 15.3 ± 3.0 ‰ in $\delta^{15}$N. In zone GL3-II, $\delta^{13}$C still varies around -23.9 ± 1.0 ‰, while $\delta^{15}$N remains steady with 10.6 ± 0.7 ‰ as it does in GL3-III with 10.3 ± 0.8 ‰, where also $\delta^{13}$C shows only little variation with -22.8 ± 0.8 ‰ (Figure 4).

The testacean assemblage of core GL3 comprises 39 taxa (Table 3). Of those, 20 belong to the moss-inhabiting hygrophilic group, including mainly the genera *Euglypha*, *Nebela* and *Pseudodifflugia*. Moss-inhabiting xerophilic testaceans mainly of





the genus *Corythion* amount to seven species. Eurybiontic species mainly of the genera *Centropyxis*, *Cyclopyxis* and *Trinema* are present with nine species. Three hydrophilic species of the genus *Difflugia* are rare and occur solely in zone GL3-I.

In zone GL3-I, varying soil moisture at the rim and the centre of the developing low-centre polygon is mirrored by xerophilic and hygrophilic species, while episodical presence of hydrophilic species indicates open-water conditions (Figure 4). Constant
wet conditions continue in zone GL3-II where hygrophilic species dominate. In zone GL3-III, xerophilic and eurybiontic species reoccur with continued presence of hygrophilic species, indicating moisture changes and occasional dry out.

The wedge ice sampled in a trench at the GL3 polygon exhibited visible vertical ice veins and numerous air bubbles < 1 mm in diameter, causing a whitish appearance. At the left edge of the exposure (Figure 3a), the wedge ice was in direct contact with the peat and clear ice above (Figure 3b). The clear ice included single vertically elongated air bubbles < 1 mm in diameter
and up to 10 mm long and very few larger air bubbles about 1 mm in diameter. Wedge ice and clear ice differ in appearance and isotopic composition. The wedge ice shows mean $\delta^{18}$O of -18.0 ± 0.8 ‰ SD, mean $\delta$D of -136.2±5.7 ‰ SD and mean *d* excess of 7.7±0.7 SD ‰ with low variation. The clear ice shows mean $\delta^{18}$O of -12.6±0.2 ‰ SD, mean $\delta$D of -98.4±1.3 ‰ and mean *d* excess of 2.2±0.8 ‰ SD. The slopes in the $\delta^{18}$O-$\delta$D cross-plot are close to GMWL for the wedge ice with 7.27 but more deviated for the clear ice with 6.09 (Figure 3c).

**4.2 Appat - core SI1 data**

The SI1 core covers the period from 5650 to -60 cal yr BP and was divided into four periods of polygon development:  SI1-I (5650 to 4670 cal yr BP, 193 to 105 cm bs), SI1-II (4670 to 3030 cal yr BP, 105 to 66 cm bs), SI1-III (3030 to 1340 cal yr BP, 66 to 48 cm bs) and SI1-IV (1340 to -60 cal yr BP, 48 to 0 cm bs) (Table 2).

The entire SI1 core is rather homogeneous and is composed of clayish grey-brown peat including single pebbles. The
20 cryostructures are not visible except for vertical ice veins about 10 cm long and < 1 mm thick at 130 to 120 cm bs and at 80 to 70 cm bs. The ice content shows some variations with a slight decrease over time and a mean value of 69±7 wt%. The active layer (10 cm thick) had a moisture content of 49 wt%.

As in the GL3 core, core SI1 shows a stepwise decrease in accumulation rates over time from SI1-I to SI1-III, while the uppermost zone, SI1-IV, returned to higher accumulation rates at 28±3 yr cm$^{-1}$ (Figure 5, Table 2).

Both TN and TOC contents exhibit an overall slight decrease over time in mean values per zone. The C/N and $\delta^{13}$C vary very little along the entire SI1 core with mean values of 12±2 and -27.9±0.4 ‰, respectively. The $\delta^{15}$N record of core SI1 reflects a decrease in mean values per zone of 2‰ between SI1-I and SI1-IV (Figure 5, Table 2).

Testaceans are rare and not diverse in core SI1 with only 12 species present (Table 3). Nine of these belong to the eurybiontic group. The wet onset of polygon evolution is reflected by hydrophilic *Arcella discoides* v. *difficilis* and hygrophilic *Nebela*
*tincta* in the lowermost zone, SI1-I, while rather dry conditions are indicated by xerophilic and eurybiontic species through most of the core (Figure 5).



## 5 Discussion

### 5.1 Regional Holocene climate history and ice wedge polygon development

The warmer-than-today climatic conditions of the Holocene Thermal Maximum (HTM) led to intense deglaciation in Arctic Canada and Greenland (Briner et al., 2016) and left large parts of the northern Baffin Bay coastlines free of glacial ice.

Radiocarbon-dated marine bivalves in raised marine deposits of the Appat study site show ages between about 9.9 and 9.2 cal yr BP (Farnsworth et al., 2018), representing the minimum age of deglaciation. This agrees well with modelled deglaciation data for the Greenland ice sheet according to which both Appat and Annikitisoq became ice free around 10 to 9 cal kyr BP (Lecavalier et al., 2014). Epigenetic freezing of the areas newly exposed by ice sheet retreat and glacio-isostatic rebound is assumed, while the setup of syngenetic permafrost aggradation as recorded in the lowermost peat zones above bedrock is dated

to 5.7 cal kyr BP  and to 4.4 cal kyr at Annikitisoq (Davidson et al., 2018). Thus, the deglaciation preceded the formation of peat and ice wedge polygon development by several thousands of years. The syngenetic permafrost onset falls into the Neoglacial cooling period with declining summer insolation after about 7.8 cal kyr BP (e.g. Briner et al., 2016; Lecavalier et al., 2017). However, the formation of ice wedge polygons are closely related to the presence of seabird colonies, which provide MDN that fertilise the vegetation and initiate peat accumulation, and the presence of breeding seabirds is in turn dependent of

the formation of the NOW polynya (Davidson et al., 2018). The polynya formation results from the development of an ice arch during spring in the southern part of Kane Basin in Nares Strait (Barber et al., 2001; Barber and Massom, 2007). The ice arch blocks the flow of drift ice from the north and the prevailing northerly winds and currents sweep the area south of the ice arch free from the new ice that continuously forms on the sea surface, keeping the polynya open. Wind-driven upwelling in the eastern part of the polynya provides heat and nutrients from deeper waters, and the combination of open water (light) and a

steady supply of nutrients to the photic zone creates the conditions for the exceptional productivity of the NOW polynya (Stirling, 1980), and in turn, the basis for the large seabird breeding populations (Møller et al., 2018).

To reconstruct Holocene temperature changes, Lasher et al. (2017) employed $\delta^{18}O$ of chironomid head capsules from Secret Lake in the Thule District as a proxy for the $\delta^{18}O$ of precipitation, which is further related to surface air temperature. This proxy approach yields maximum estimates of Holocene temperature changes but is, as the study states, summer- and early

autumn-biased. The inferred summer season temperatures that were up to 4 °C warmer than today decreased from about 7.7 cal kyr BP until about 2.3 cal kyr BP before reaching colder than today temperatures, including the coldest period after about 1.2 cal kyr BP (Lasher et al., 2017). The reconstructed period of decreasing summer temperatures covers the onset of permafrost aggradation at both sites, on Appat and at Annikitisoq, and likely relates to the dynamics of the NOW polynya as reflected in sea surface temperature (SST), sea surface salinity (SSS) and sea ice cover (SIC) proxy data from marine sediments

such as dinocyst records (Levac et al., 2001). After the breakup of perennial sea ice cover in the northern Baffin Bay around 10.5 cal kyr BP, Holocene minima in SIC with up to 4-5 ice-free months per year occurred between about 7.4 to 4 cal kyr BP accompanied by maxima August SST and SSS (Levac et al., 2001). The same interval is represented in benthic foraminifera




records of species composition and δ¹³C, indicating occasional upwelling between 7.3 and 4 cal kyr BP and maximum upwelling between 4 and 3.1 cal kyr BP (Knudsen et al., 2008).

The exceptional setting of climate and oceanographic conditions maintained the polynya that attracted seabirds to colonise the shores of the northern Baffin Bay and controlled, by doing so, the onset of peat accumulation and syngenetic ice wedge polygon

growth. The spatial distribution of the polygonal peat development is related to seabird colonies. Nonetheless, the birds' control of peat and thus permafrost accumulation is further altered by internal polygon dynamics that are traceable by testacean records (Bobrov et al., 2013). The onset of polygon formation in rapidly aggrading peat above bedrock since 4400 cal yr BP is likely indicated by the presence of xerophilic *Corythion* species in the lowermost part of GL3-I. These species build their shells from silicate, and their presence might thus point to bedrock weathering during initial polygon mire formation. The change to

stagnant water in a low-centre polygon is mirrored by the occurrence of hydrophilic *Difflugia* species within zone GL3-I. The even presence of hygrophilic species such as *Nebela tincta* and *Pseudodifflugia gracilis* in zones GL3-I and GL3-II further mirrors constantly wet conditions during the formation of these zones until about 2620 cal yr BP. In zone GL3-III, xerophilic species such as *Assulina muscorum* and *Corythion dubium* point to varying moisture conditions with drier periods. The latter likely indicates dry-up of degrading polygons when low-centre polygons transform into high-centre polygons due to starting

wedge ice meltout and concurrent inversion of the polygonal micro-topography during the GL3-III deposition. Today, the polygons are present in their degradation stages as high-centre polygons when the initially grown ice wedges melt out and form trenches between the polygon centres.

An ice wedge remnant was sampled at Annikitisoq (Figure 3) at about 1.2 m depth below the GL3 polygon surface. This position suggests that the wedge ice formed mainly during accumulation of the GL3-II core zone, i.e. after 3520 cal yr BP

although direct dating of the ice is not available. The wedge ice shows low variation in δ¹⁸O, δD, and deuterium excess. The slope of 7.27 of the wedge ice data fits that of the modern Thule (Local) Meteoric Waterline (7.33; Figure 3c) indicating the meteoric moisture source of the wedge ice. The offset between mean values of the modern cold season (Oct-May) precipitation in Thule (-27 ‰ in δ¹⁸O, -207 ‰ in δD, GNIP database, 2018) and the Holocene wedge ice (-18 ‰ in δ¹⁸O, -136 ‰ in δD) data is obvious. Comparable observations are known from studies in Siberia (e.g. Opel et al., 2011) and Canada (e.g. Grinter

et al., 2018) and explained by isotopic changes within the snow cover prior to melting (Opel et al., 2018) as well as exchange between liquid water, snow and ice during snowmelt and re-freezing. Thus, the wedge ice data is seen as the late Holocene cold season precipitation signal. In contrast, the water building up the clear ice underwent evaporation before freezing as shown by the lower slope of 6.09 (Figure 3c). The clear ice likely originated from summertime runoff water that ponded in the trench above the degrading ice wedge. It shows values of about -13 ‰ in δ¹⁸O and -98 ‰ in δD, which if compared with warm season

precipitation (Jun-Sept, -19 ‰ in δ¹⁸O and -142 ‰ in δD, GNIP database, 2018) is isotopically heavier, most likely due to evaporation (Figure 3c). The late Holocene wedge ice data from Annikitisoq are the first of such records from Greenland, while comparable stable water isotope records of the High Arctic are rare and limited to Svalbard (Vasil'chuk et al., 2015; Vittinghus et al., 2008).



In summary, the Holocene NOW polynya dynamics controlled by oceanographic and climate variation enabled bird colonisation of shores of the Thule District. The recorded onset of bird presence and thus of ice wedge polygon development at about 4400 cal yr BP at Annikitisoq and at about 5650 cal yr BP on Appat (Davidson et al., 2018) falls within the period of decreasing temperatures and decreasing SIC, and SST and SSS maxima and the deduced establishment of the NOW polynya

(Levac et al., 2011). Thus, the syngenetic permafrost formation directly depending on peat deposition (controlled by bird activity) indirectly follows the Holocene climate trends.

## 5.2 Organic matter signatures as reflection of bird presence

A study by Zwolicki et al. (2013) compared the impact of a little auk and a mixed thick-billed murre and kittiwake (*Rissa tridactyla*) colony on soil physical and chemical properties at the northern coast of Hornsund Fjord, Spitsbergen. The study

emphasises that bird presence generally enhances nutrient content and soil conductivity and lowers pH. However, the different food sources of the birds, i.e. planktivorous little auk vs. piscivorous thick-billed murre and kittiwake, affect the colony surroundings in different ways. The main differences between the little auk colony at Annikitisoq and the thick-billed murre colony on Appat are apparent in TOC, TN and C/N data in the uppermost active layer samples of the respective core units GL3-III and SI1-IV. They comprise about twofold higher TOC and twofold lower TN in GL3-III active layer (with little auk)

compared with the SI1-IV active layer (thick-billed murre), resulting in a large difference in C/N (40 in GL3-III compared to 17 in SI1-IV).

The OM input from bird colonies with $^{15}$N enriched isotopic composition (i.e. higher $\delta^{15}$N) is traceable in the developing peat below the colonies (González-Bergonzoni et al., 2017; Gąsiorowski and Sienkiewicz, 2019). The magnitude of the little auk's fertilising influence on the colony surroundings can be considerable, ranging from 1.2 g m$^{-2}$ direct deposition of dry faeces per

20 day in a little auk colony in Spitsbergen to 0.5 g m$^{-2}$ per day at a distance of 200 m from the bird colony towards the sea (Zwolicki et al., 2013). In the NOW area, little auks are estimated to deposit >3500 tons of N in the terrestrial environment each year, and to affect vegetation productivity over an area of >200 km$^2$ (Mosbech et al. 2018). If the $\delta^{15}$N signature of peat below colonies is taken as a rough proxy of the productivity of the colony, the peat accumulation rates and $\delta^{15}$N should relate to each other. In the GL3 core from Annikitisoq, three zones can be differentiated with accumulation rates decreasing from

25 5±1 yr cm$^{-1}$ in GL3-I to 13±3 yr cm$^{-1}$ in GL3-II and 38±12 yr cm$^{-1}$ in GL3-III, while the $\delta^{15}$N becomes more depleted from 15.3±3.0‰ in GL3-I to 10.6±0.7‰ in GL3-II and 10.3±0.8‰ in GL3-III (Table 2). The ice content of the frozen peat in the GL3 core constitutes most of the material and is comparably high for all three units, the only decrease occurring in the uppermost part of unit GL3-III (Figure 4). Thus, the variation in the accumulation rate is related to past vegetation growth and peat formation rather than to permafrost (ice) aggradation. The relation between $\delta^{15}$N as a tracer of colony productivity and

30 peat accumulation is not linear; it undoubtedly further depends on the N turnover and isotopic fractionation along the pathway from source (food) to faeces via the birds' metabolism, further to vegetation and, finally, to the peat via microbial metabolism. However, the thick-billed murre core SI1 from Appat shows no relation between peat accumulation rates, which are generally





lower than the GL3 core, and OM signatures, which are much less variable than the GL3 core (Figure 5, Table 2). The difference in peat accumulation rates between the sites might relate to the different preferences of the bird species for breeding. While the little auk commonly breeds on the coastal fringe in screes continuing somewhat inland (Mosbech et al., 2018), most thick-billed murre colonies are steep cliffs directly facing the coast (Kampp, 1990). Thus MDN input of the latter on

terrestrial colony surroundings is reduced if compared with those of little auk.

When comparing the C/N ratios with $\delta^{13}$C, both sites exhibit a clear marine source of the bird-derived OM as seen in similar signatures in food and faeces for little auk and in faeces of thick-billed murre (Figure 6). Both peat cores show clear terrestrial signatures in the C/N-$\delta^{13}$C relation, but the GL3 core shows a large shift from the marine source in C/N at relatively stable $\delta^{13}$C (Figure 6a), while the SI1 core shows a large shift in $\delta^{13}$C at relatively stable but low C/N (Figure 6b).

This pattern reflects the temporal dimension of OM decomposition when high accumulation rates as observed in the GL3 core ($12\pm13$ yr cm$^{-1}$) induce less time for decomposition in the uppermost active layer before becoming permafrost. This is seen in high C/N (core mean of $47.5\pm12.7$), indicating a low degree of decomposition (Kuhry and Vitt, 1996) and almost unaltered $\delta^{13}$C. Compared with the food and faeces signatures, the latter shows only a small variation of about 1‰ between the mean $\delta^{13}$C of -24‰ in both food and faeces and the mean $\delta^{13}$C of $-23.1\pm1.2$‰ in the GL3 core (Figure 6a). In contrast, the SI1 core

shows more than two-fold lower accumulation rate (core mean of $30\pm26$ yr cm$^{-1}$) than core GL3 (core mean of $12.3\pm2.0$ yr cm$^{-1}$) and low C/N (core mean of $12.3\pm2.0$) indicating a higher degree of OM decomposition if compared with the GL3 core. The $\delta^{13}$C source signal ($\delta^{13}$C of -22.2‰) is over 5‰ less depleted than the SI1 peat (core mean $\delta^{13}$C of $-27.9\pm0.4$‰), which further support a stronger OM alternation in the SI1 peat than in the GL3 peat (Figure 6b).

Nitrogen distribution and turnover influenced by little auk have been studied at the northern shore of Hornsund Fjord in

Spitsbergen (Skrzypek et al., 2015). There, the major N sources have been examined by means of $\delta^{15}$N and quantified as atmospheric deposition (38 %), primary N$_2$-fixation from the atmosphere (26 %) and bird faeces deposition (36 %). Thus, the impact of bird presence is notable as also shown by González-Bergonzoni et al. (2017) in samples from NW Greenland. While tracking the pathway of $\delta^{15}$N from the marine realm to land via little auks, Skrzypek et al. (2015) found similar signatures in the birds' food of $7.0\pm0.9$‰ (Wold et al., 2011) and in the faeces of $8.1\pm0.5$‰. In the little auk data from Annikitisoq, the shift

from food ($\delta^{15}$N of $7.3\pm0.0$‰) to faeces ($\delta^{15}$N of $6.2\pm1.0$‰) is comparably small and almost within the standard deviation ranges of each other. Substantial parts of the N provided by bird presence likely spread in the area by outwash after snowmelt and become altered by microbial turnover (ammonia volatilisation), resulting in $^{15}$N enriched substrate with a comparably heavier $\delta^{15}$N composition (Skrzypek et al., 2015).

The microbial-biased fractionation from bird faeces to plant available substrate amounts to about 5‰ (Figure 7) and is seen in

the isotopic composition of the uppermost sample ($\delta^{15}$N of 10.8‰ at 0-5 cm bs) of the GL3 core. We assume that the GL3 core is affected both by numerous droppings on the surface, as observed, and additional outwash from the colony situated close by upslope from the coring site, which is a similar setting as in the Ornithocoprophilous and wet moss tundra types studied by Skrzypek et al. (2015) at Hornsund, Spitsbergen. There, the share of bird-related N input used by plants reaches up to 100%. The $\delta^{15}$N of *Polytrichastrum alpinum* moss, which is present in the GL3 core, reaches values between 9.2 and 11.4‰ in the




Ornithocoprophilous tundra (bulk soil 11.4±1.4‰) and between 10.1 and 10.8‰ in the wet moss tundra (bulk soil 8.8±3.4‰ to 13.1±1‰). Thus, the shift in the Spitsbergen data of up to 4‰ between little auk faeces and moss and up to 5‰ between little auk faeces and bulk soil is of the same extent as that demonstrated from the Annikitisoq data.

Comparable stable isotope data of piscivorous thick-billed murre food has not yet been obtained. The single information presented here shows no shift between faeces ($\delta^{15}$N of 5.1‰) and surface ($\delta^{15}$N of 5.1‰ at 0-10 cm bs) and a shift of about 7‰ between faeces and the uppermost zone SI1-IV (mean $\delta^{15}$N of 12.4±2.8‰, Table 2). It is notable in both the GL3 and SI1 cores that the mean $\delta^{15}$N per zone decreases bottom-up, i.e. from old via young to modern sources (Figure 7, Table 2).

In summary, impact of seabird colony presence is evidenced for both studied species and these results are comparable with previous studies (e.g. González-Bergonzoni et al., 2017; Skrzypek et al., 2015; Zwolicki et al., 2013). The $\delta^{15}$N signature of permafrost OM preserves the bird-induced MDN input and might serve as a tracer of colony size over time, although $\delta^{15}$N fractionation by microbial turnover alters the signal by about 5‰ in the little auk-related data and by about 7‰ in the thick-billed murre-related data from bird faeces to plant available substrate. Peat accumulation rates driven by vegetation growth correspond to the observed $\delta^{15}$N trends where decreasing peat accumulation leads to increasing depletion of $\delta^{15}$N as seen in the little auk-related data of core GL3, while thick-billed murre-related data of core SI1 show more than two-fold lower accumulation rates, a much higher degree of OM decomposition and an almost stable $\delta^{15}$N.

## 6 Conclusions

The impact of seabird colonies on High Arctic terrestrial environments is seen in the input of marine-derived nutrients that is traceable by geochemical signatures of organic matter in C/N to $\delta^{13}$C relations. The resulting fertilisation facilitated late Holocene vegetation growth and subsequent peat accumulation and permafrost aggradation in otherwise barren areas. In the study area in NW Greenland, ice wedge polygons developed during the late Holocene in the vicinity of both planktivorous little auk and mainly piscivorous thick-billed murre colonies. The two bird species affect the surroundings of their colonies in different ways as reflected by the about twofold higher TOC and twofold lower TN in the uppermost samples of the little auk-affected peat compared with the samples of thick-billed murre-affected peat, resulting in a large difference in C/N. The little auk-affected peat core shows decreasing accumulation rates and decreasing $\delta^{15}$N (in mean values per core zone) from old to young, while variations in accumulation rates of thick-billed murre-affected peat are not concurrent with the almost stable $\delta^{15}$N. The concurrent onset of sea bird colonies and syngenetic ice wedge polygon formation after the establishment of the NOW polynya during the Neoglacial cooling highlights the complex response of biologic and permafrost dynamics to climatic and oceanographic conditions of the late Holocene.




## Author contributions

AM and TAD initiated and designed the present study. SW, TAD and IGB drilled and documented the cores supported by KLJ and AM. TW and IGB carried out stable isotope and OM analyses. AB carried out testacean analysis and interpretation. TO and EJ supported the overall data analysis and interpretation. SW and TAD wrote the paper with contributions from the other co-authors, who contributed equally to the final discussion of the results and interpretations.

## Competing interests

The authors declare that they have no conflict of interest.

## Acknowledgements

This study is part of The North Water Project (www.NOW.KU.DK) funded by the Velux Foundations and the Carlsberg Foundation of Denmark. The base commander at Thule Air Base kindly gave us permission to use the base for transit, while the Danish Liaison Officer helped with arranging our stay. We are indebted to Tony Rønne Pedersen, Greenland Contractors (GC), for extensive logistic and practical help, and to Erland Søndergård (GC) and Hans Otzen (GC) for boat transport on Blue Jay and Hot Totty. We are also very grateful to the communities in Qaanaaq and Siorapaluk for their hospitality. TO and SW acknowledge funding from the German Research Foundation (grants OP217/4-1 and WE4390/7-1, respectively). Laboratory analyses were expertly undertaken by Hanno Meyer, Mikaela Weiner and Dyke Scheidemann (AWI Potsdam). We thank Anne Mette Poulsen (Aarhus University) for manuscript editing.

## Data availability

Original data are available at https://www.pangaea.de after acceptance of this paper.

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



**Tables**

Table 1: Cryolithological description of peat core GL3 and wedge ice from Annikitisoq and peat core SI1 from Appat in the Thule district (NW Greenland).

| ID | Material | Depth [cm bs] | № of samples | Description | Cryostructures |
|---|---|---|---|---|---|
| **Annikitisoq (Great Lake), 01-AUG-15, 76.03288°N, 67.61811°W** | | | | | |
| GL3 -IW | wedge ice exposure | 50 to 80 orizontal profiles; spanning 1.7 m in total | 35 | syngenetic ice wedge at GL3-3 polygon, expressed shoulders, degrading from above, in trench between two high-centre polygons | **ice wedge ice:** whitish, vertical ice veins and numerous air bubbles ø < 1 mm; **clear ice:** clear, single vertically elongated air bubbles ø < 1 mm and up to 10 mm long, and very few larger air bubbles ø 1 mm |
| **Annikitisoq (Great Lake), 31-JUL-15, 76.03288°N, 67.61811°W** | | | | | |
| GL3 | pit, high-centre polygon | 0 to 20 | 5 | **0->20 cm bs:** light-brown to brown moss peat, low decomposition, no mineral content | thawed active layer |
| GL3 | core, high-centre polygon | 20 to 320 | 70 | **20->130 cm bs:** brown moss peat, low decomposition, no mineral content; **130->320 cm bs:** reddish light-brown partly stratified peat | non-parallel, wavy lenticular and irregular reticulated, ice lenses > 1mm thick |
| **Appat (Saunders Ø), 27-JUL-15, 76.56908°N, 70.04099°W** | | | | | |
| SI1 | pit, high-centre polygon | 0 to 10 | 1 | **0->10 cm bs:** brown peat, living roots, visible plant remains, low decomposition, pebbles | thawed active layer |
| SI1 | core, high-centre polygon | 10 to 198 | 48 | **10->198 cm bs:** brown peat, visible plant remains, pebbles | lenticular, ice lenses < 1 mm thick |



Table 2: Modern reference and core data from Annikitisoq and Appat showing the core zonation by depth and time intervals, the accumulation rates as well as the mean values and standard deviation of organic matter properties per zone and per entire core.

| | Depth [cm bs] | Age [cal yr BP] | Rate [yr cm$^{-1}$] | TOC [wt%] | TN [wt%] | C/N | $\delta^{13}$C [‰] | $\delta^{15}$N [‰] | n |
|---|---|---|---|---|---|---|---|---|---|
| **Annikitisoq** *modern little auk reference* | | | | | | | | | |
| food | 0 | modern | -- | 46.1 ± 0.0 | 7.9 ± 0.0 | 6.8 ± 0.0 | -24.1 ± 0.0 | 7.3 ± 0.0 | 1 |
| faeces | 0 | modern | -- | 32.2 ± 6.8 | 6.9 ± 1.7 | 5.6 ± 1.4 | -24.0 ± 0.8 | 6.2 ± 1.0 | 4 |
| **Annikitisoq** *core GL3* | | | | | | | | | |
| GL3-III | 58 - 0 | 2620 to 540 | 38 ± 12 | 46.5 ± 0.6 | 1.0 ± 0.3 | 47.5 ± 10.4 | -22.7 ± 0.8 | 10.3 ± 0.8 | 12 |
| GL3-II | 129 - 58 | 3520 to 2620 | 13 ± 3 | 46.9 ± 0.9 | 0.9 ± 0.1 | 50.8 ± 10.1 | -23.9 ± 1.0 | 10.6 ± 0.7 | 18 |
| GL3-I | 320 - 129 | 4400 to 3520 | 5 ± 1 | 44.0 ± 1.4 | 1.0 ± 0.3 | 46.2 ± 14.1 | -23.0 ± 1.3 | 15.3 ± 3.0 | 45 |
| all GL zones | 320 - 0 | 440 to 540 | 12 ± 13 | 45.1 ± 1.8 | 1.0 ± 0.3 | 47.5 ± 12.7 | -23.1 ± 1.2 | 13.4 ± 3.4 | 75 |
| **Appat** *modern thick-billed murre reference* | | | | | | | | | |
| **faeces** | 0 | modern | -- | 24.9 ± 0.0 | 2.0 ± 0.0 | 8.2 ± 0.0 | -22.2 ± 0.0 | 5.1 ± 0.0 | 1 |
| **Appat** *core SI1* | | | | | | | | | |
| SI1-IV | 48 - 0 | 1340 to -60 | 28 ± 3 | 23.9 ± 3.6 | 2.2 ± 0.3 | 11.2 ± 1.8 | -27.9 ± 0.5 | 12.4 ± 2.8 | 11 |
| SI1-III | 66 - 48 | 3030 to 1340 | 95 ± 16 | 25.6 ± 1.7 | 2.3 ± 0.2 | 11.1 ± 0.2 | -28.2 ± 0.1 | 13.3 ± 1.4 | 10 |
| SI1-II | 105 - 66 | 4670 to 3030 | 43 ± 5 | 29.2 ± 7.4 | 2.5 ± 0.5 | 11.7 ± 1.1 | -27.9 ± 0.3 | 14.2 ± 0.4 | 5 |
| SI1-I | 193 - 105 | 5650 to 4670 | 11 ± 1 | 38.2 ± 6.6 | 2.9 ± 0.4 | 13.1 ± 2.0 | -27.8 ± 0.5 | 14.4 ± 0.7 | 23 |
| all SI zones | 193 - 0 | 5650 to -60 | 30 ± 26 | 32.1 ± 8.6 | 2.6 ± 0.5 | 12.3 ± 2.0 | -27.9 ± 0.4 | 13.8 ± 1.7 | 49 |





Table 3: Presence of testacean species and their ecological indication in core GL3 from Annikitisoq and core SI1 from Appat (Saunders Ø).

| | core GL3 | core SI1 |
|---|:---:|:---:|
| **eurybiontic species** | | |
| *Centropyxis aerophila* | x | x |
| *Centropyxis aerophila* v. *minuta* | | x |
| *Centropyxis constricta* | x | x |
| *Centropyxis* v. *minima* | x | x |
| *Centropyxis sylvatica* | | x |
| *Centropyxis sylvatica* v. *minor* | | x |
| *Cyclopyxis eurystoma* | x | x |
| *Cyclopyxis eurystoma* v. *parvula* | | x |
| *Euglypha laevis* | x | |
| *Plagiopyxis* cf. *callida* | | |
| *Schoenbornia humicola* | x | |
| *Trinema complanatum* | x | |
| *Trinema lineare* | x | x |
| *Trinema lineare* v. *terricola* | x | |
| **moss-xerophilic species** | | |
| *Arcella arenaria* v. *compressa* | x | x |
| *Assulina muscorum* | x | |
| *Valkanovia elegans* | x | |
| *Corythion dubium* | x | |
| *Corythion dubium* v. *minima* | x | |
| *Corythion dubium* v. *orbicularis* | x | |
| *Corythion pulchellum* | x | |
| **moss-hygrophilic species** | | |
| *Centropyxis elongata* | x | |
| *Euglypha aspera* | x | |
| *Euglypha compressa* f. *glabra* | x | |
| *Euglypha* cf. *denticulata* | x | |
| *Euglypha strigosa* | x | |
| *Euglypha strigosa* f. *glabra* | x | |
| *Nebela collaris* | x | |
| *Nebela lageniformis* | x | |
| *Nebela parvula* | x | |
| *Nebela tincta* | x | x |
| *Pseudodifflugia gracilis* | x | |
| *Pseudodifflugia gracilis* v. *terricola* | x | |
| *Pseudodifflugia fascicularis* | x | |
| *Pseudodifflugia* cf. *horrida* | x | |
| *Pseudodifflugia fulva* | x | |
| *Pseudodifflugia jungi* | x | |
| *Pseudodifflugia* sp. 1 | x | |
| *Pseudodifflugia* sp. 2 | x | |
| *Trinema complanatum* v. *elongata* | x | |
| *Trinema complanatum* v. *platystoma* | x | |
| **hydrophilic species** | | |
| *Arcella discoides* v. *difficilis* | | x |
| *Difflugia bacillariarum* | x | |
| *Difflugia lucida* | x | |
| *Difflugia pristis* | x | |



**Figures**

Figure 1: The study area in the Thule District, NW Greenland, including the location of coring sites and the distribution of breeding colonies of little auk and thick-billed murre after Boertmann and Mosbech (1998) and The Greenland Seabird Colony Register, maintained by Danish Center for Environment and Energy, Aarhus University, and Greenland Institute of Natural Resources.







Figure 2: High-centre polygons at Annikitisoq (a-c) and on Appat (d-e). Coring sites are shown in (c) and (e). Note persons for scale.





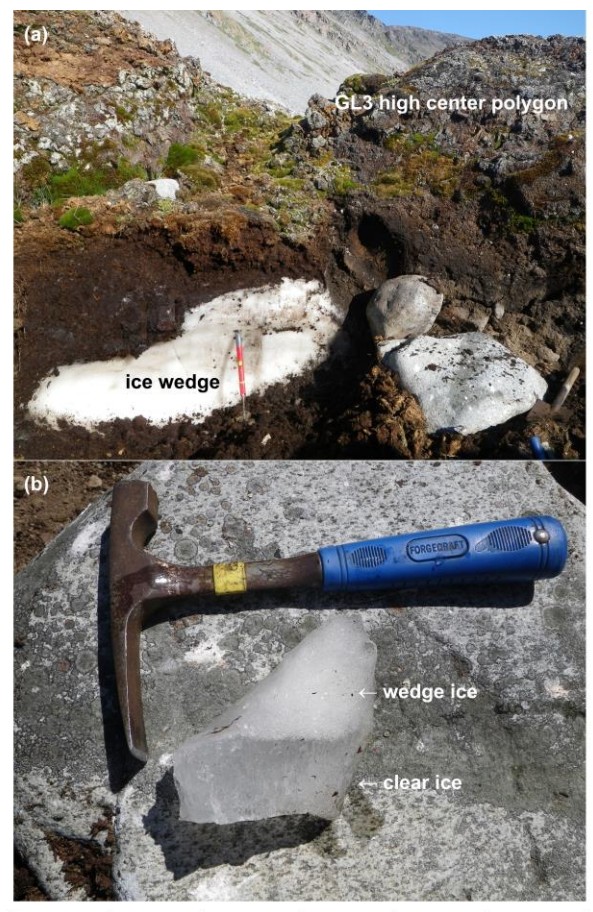

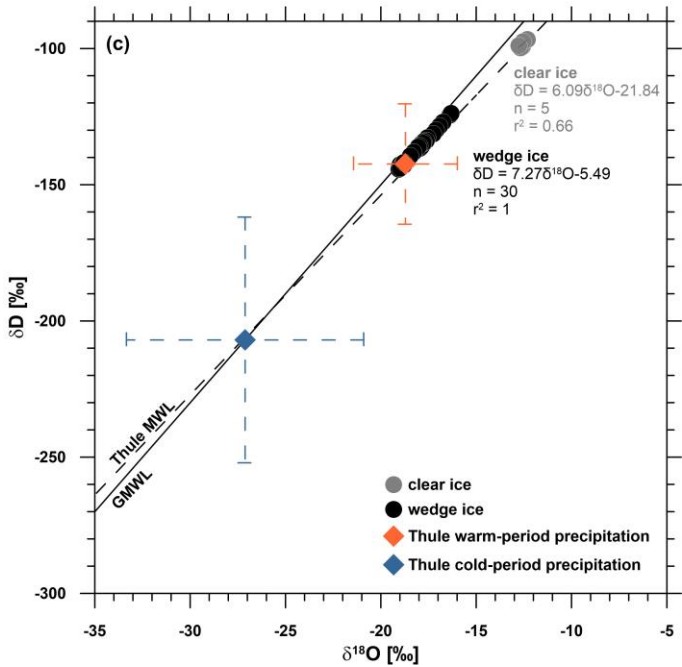

Figure 3: Ground ice exposed in a trench at the GL3 polygon (a) including wedge ice and clear ice (b), which differ in their isotopic composition (c). Co-isotopic data are shown along with the Global Meteoric Water Line (GMWL; Craig, 1961) and the Thule MWL (GNIP database, 2018) based on precipitation data from 1966-1970 (monthly means, n=43, $\delta D = 7.33\delta^{18}O–7.14$, $R^2=0.97$). The mean modern cold season (October to May, n = 29) and warm season (June to September, n = 14) precipitation data and standard deviation are shown by blue and orange diamond symbols, respectively (GNIP database, 2018).





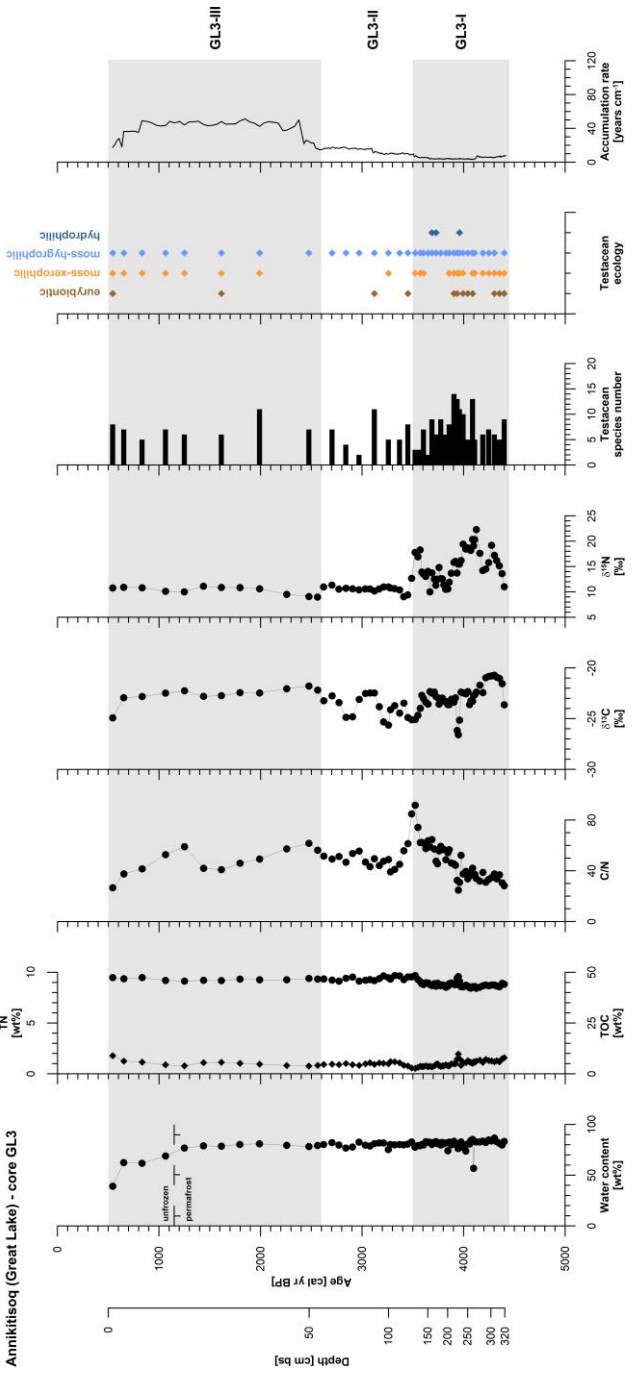

Figure 4: OM data and age information according to Davidson et al. (2018) as well as testacean ecological groups and zonation of core GL3 from Annikitisoq. TN data are given as black diamonds and refer to the upper x-axis.





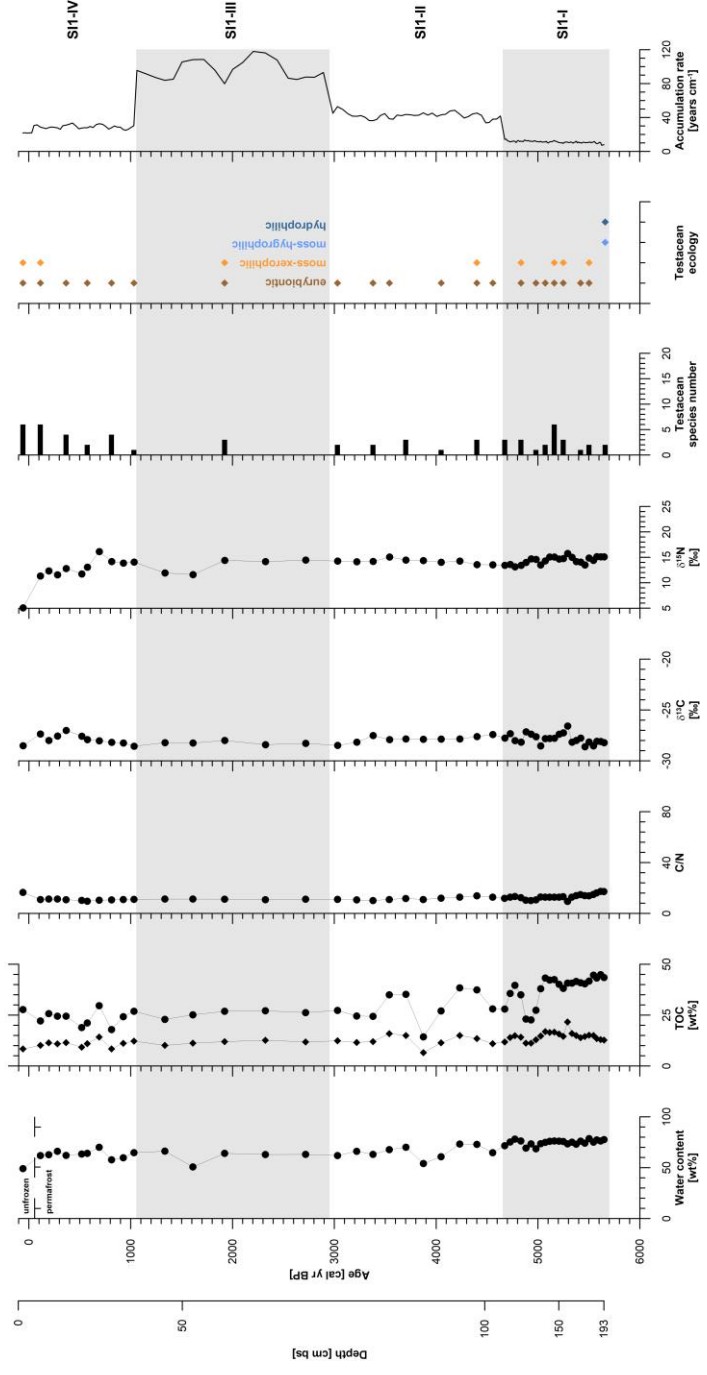

Figure 5: OM data and age information according to Davidson et al. (2018) as well as testacean ecological groups and zonation of core SI1 from Appat. TN data are given as black diamonds and refer to the upper x-axis.





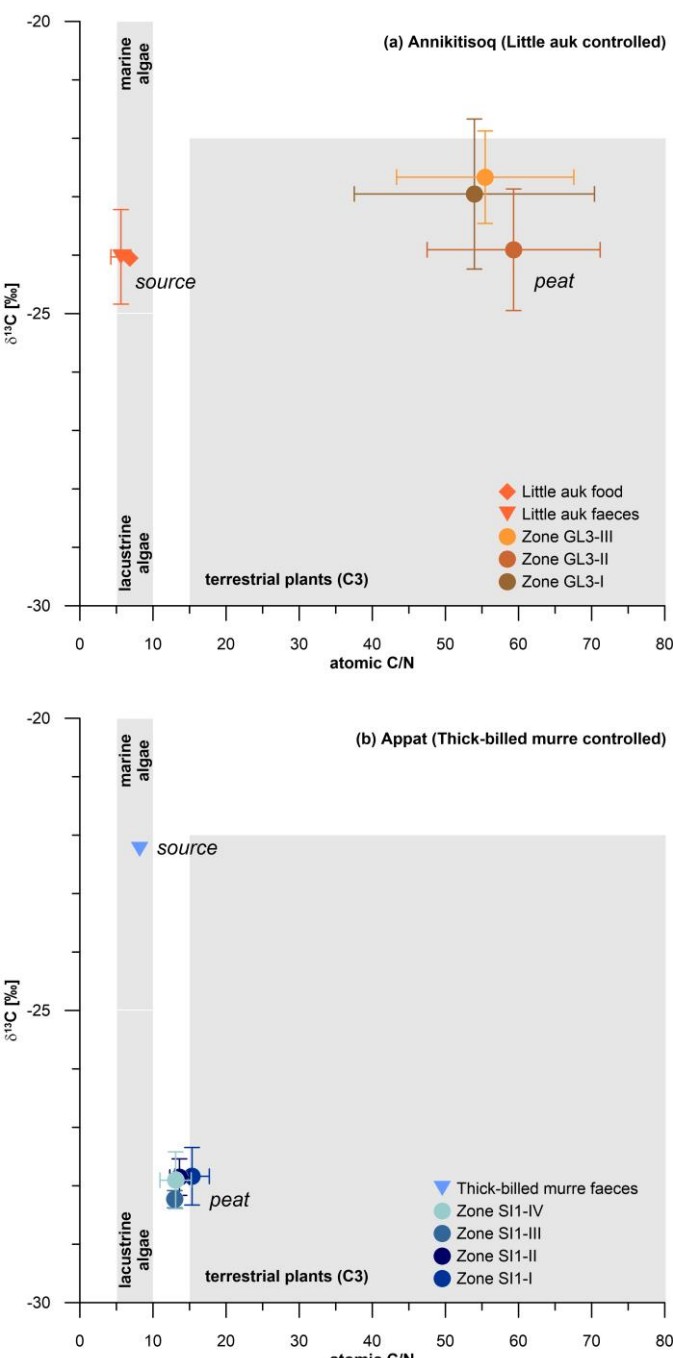

Figure 6: C/N and $\delta^{13}$C composition of bird-controlled permafrost areas on (a) Annikitisoq and (b) Appat compiling data of MDN sources such as bird food (diamonds) and bird faeces (triangles), and of the peat core zones of GL3 and SI1 (circles).

5   Data are given in Table 2. Grey shaded areas refer to Meyers (1997).



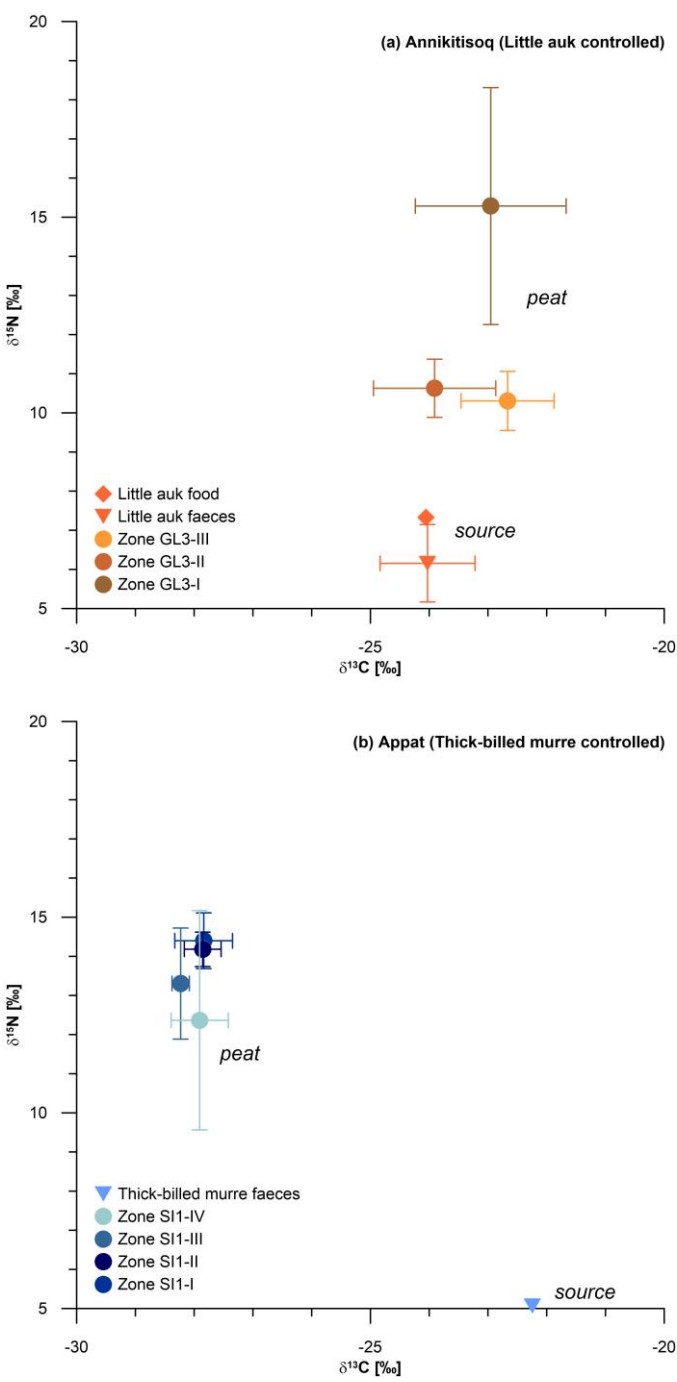

Figure 7: δ¹³C and δ¹⁵N composition of bird-controlled permafrost areas on (a) Annikitisoq and (b) Appat compiling data of MDN sources such as bird food (diamonds) and bird faeces (triangles), and of the peat core zones of GL3 and SI1 (circles).

5    Data are given in Table 2.