# Peer review of "Stable isotope signatures of Holocene syngenetic permafrost trace seabird presence in the Thule District (NW Greenland)"

_Biogeosciences, 2019_

## Referee Comment (RC1) · Anonymous Referee #1 · 30 Apr 2019

Review of ms 'bg-2019-71' submitted to Biogeosciences journal. The manuscript by Wetterich et al. is an interesting attempt of evaluation of the role of seabird as a factor controlling Arctic tundra development. It try to link seabird colonization with climate change over the last 5,000 yrs. The manuscript is well written, easy to read and understand. However, I would ask for some more explanation and small correction, mainly concerning sediment dating and presentation of analytical results.

1. The radiocarbon dating and age-depth modeling could be describe more detail. I know, that it was a topic of paper by Davidson et al. 2018, but still I have some doubts:
- Some samples for dating were collected from significantly thick core segments (even

10 cm long for topmost part of SI 1). Do author consider depth uncertinity in age-depth models? It can be significant. - Please give some details on geological settings. According to my knowlage, at least in Saunders Is there is a lot of carbonate rocks. It could substantially impact radiocarbon dating ('old carbon effect), since bulk sediment/peat samples were dated. How did the authors did overcome this issue? - What sediment/rocks were in the cores' basement? Are peat profile represent entire biogenic sedimentation on the spots? 2. How were the cores divided into zones/periods? What were the criteria? Was the division evaluated with statistical analysis (is it significant?) 3. Please give more detail on subfossil Testacea analysis methodology, namely how big samples were taken to analysis, counted quantities etc. I would see 'classical' percentage diagrams rather. 4. The values presented in the Table 2 (results for food and faeces) cannot have uncertainty 0.0. Please report at least measurement uncertainties. I hope, that in means calculation authors include measurement uncertainty and applied 'error propagation rule' for mean uncertainty calculations.

Small corrections: - Table 2: in row all GL zones should be '4400 to 540' - Page 1, row11: 'ground at sea' sounds tricky, consider change to 'area' - Page 5, row2: please report accuracy in ‰

---

## Referee Comment (RC2) · Anonymous Referee #2 · 6 Jul 2019

Comments on manuscript by Wetterich et al. submitted to Biogeosciences. The manuscript is well written and fairly well illustrated and the subject is suitable for Biogeosciences. However, the authors need to refer to previous work carried out in the area. Over 30 radiocarbon ages from peat deposits below sea-bird colonies from the Thule region were reported by Delibrias et al. (1972) and discussed by Malaurie et al. (1972). The peat deposits were also discussed by Salomonsen (1979). The authors could also refer to a paper by Burnham et al. (2009). Malaurie et al. discussed peat accumulation rates – an issue also discussed by Wetterich et al.

The oldest ages reported by Malaurie were ca. 1500 years BP. Wetterich et al. have

ages going back to ca. 5000 years. I wonder if the lack of older ages is due to the sampling methods? I would assume that sea birds colonised NW Greenland already in the Early Holocene, but I agree that colonies expanded during the Late Holocene. We know from radiocarbon dating of shells of marine molluscs in the region that there was a rich marine invertebrate fauna already in the Early Holocene. And of course there are also very large sea-bird colonies in other parts of Greenland.

The authors describe some of the cores as brown moss peat. I wonder if the mosses were identified? The peat on Nordvestø consists of Aplodon wormskioldii, which is a most uncommon fossil species. I wonder if the other peat deposits in the Thule region are also dominated by this species? On page 10, line 35 we hear about Polytrichastrum alpinum, which is present in the GL3 core. Does it dominate?

Some other comments Page 2, line 8. How many pairs of thick-billed murre? Page 2, line 33 and 34. I prefer use rather than employ Page 3, line 18. Please provide data on the elevation of the sites. Are the sites below or above the marine limit? Page 3, line 24. Please provide data on the local vegetation as you observed in the field. Grasses and sedges are much more important than rushes in the area. Not sure about dwarf shrubs at the actual sampling sites. Page 4, line 2. How do you know you reached bedrock? Could it be stones or boulders? From the photos it appears that both sampling sites are located in areas with thick gravel deposits. Page 5, line 4-10. More information on testacean analysis is needed. How many specimens were counted per sample? What was the sample size? It would be nice to see some testacean diagrams. Page 5, line 14. The succession was divided into three zones Page 5, line 28. Is there only one testacean assemblage in the whole core? Page 7, line 3-4. Intense deglaciation occurred in the Early Holocene, long before the Holocene Thermal Maximum – due to abrupt warming at the Younger Dryas-Holocene transition. With respect to the deglaciation chronology see Bennike & Björck (2002). Page 7, line 5 9.9 and 9.2 kyr BP – not yr BP! Page 7, line 3-8. According to the compilation by Bennike & Björck (2002) the two study sites were deglaciated before 10 cal. kyr BP

Page 11, line 23. I think the main difference is the number of little auks compared with thick-billed murre. Also the different nesting places as mentioned by the authors.

Some references: Bennike, O. & Björck, S. 2002: Chronology of the last recession of the Greenland Ice Sheet. Journal of Quaternary Science 17, 211–217. Burnham, K.K., Burnham, W.A. & Newton, I. 2009: Gyrfalcon Falco rusticolus post-glacial colonization and extreme long-term use of nest-sites in Greenland. Ibis 151, 514–522. Delibrias, G., Guillier, M.T. & Labeyrie, J. 1972: Gif natural radiocarbon measurements VII. Radiocarbon 14, 280–320. Malaurie J, Vasari Y, Hyva ÂÍrinen H, Delibrias G, Labeyrie J (1972) Preliminary remarks on Holocene paleoclimates in the regions of Thule and Inglefield Land, above all since the beginning of our era. Acta Universitatis Ouluensis, Series A, Scientiae Rerum Naturalium 3, Geologica 1, 105–136. Salomonsen F (1979) Ornithological and ecological studies in southwest Greenland (59°46'–62°27'N. Lat). Meddelelser om Grønland 204(6), 214 pp.

---

## Author Comment (AC1) · 12 Sep 2019

Responses from the authors to comments of anonymous Referee #1 - bg-2019-71

The manuscript by Wetterich et al. is an interesting attempt of evaluation of the role of seabird as a factor controlling Arctic tundra development. It try to link seabird colonization with climate change over the last 5,000 yrs. The manuscript is well written, easy to read and understand. However, I would ask for some more explanation and small correction, mainly concerning sediment dating and presentation of analytical results. REPLY: We are grateful to referee #1 for the time and effort spent on reviewing our manuscript.

[Figure]

1. The radiocarbon dating and age-depth modeling could be describe more detail. I know, that it was a topic of paper by Davidson et al. 2018, but still I have some doubts: Some samples for dating were collected from significantly thick core segments (even 10 cm long for topmost part of SI 1). Do author consider depth uncertinity in age-depth models? It can be significant.

REPLY: All dated samples derive from high-resolution sampling of 4- to 5-cm thick core segments given the amount of material needed for the various analyses. The only exception is the mentioned sample from the uppermost SI1 core that represents the 10-cm-thick active layer with an expected recent age of 51 cal yr BP. Here, in the uppermost thawed cover, more detailed subsampling seems inappropriate. Therefore depth uncertainty in the age-depth models is of minor relevance for the present core records. The complete presentation of the age-depth models from Davidson et al. (2018) can be found as Supplementary Material here: https://static-content.springer.com/esm/art%3A10.1007%2Fs13280-018-1031-1/MediaObjects/13280_2018_1031_MOESM1_ESM.pdf

Please give some details on geological settings. According to my knowlage, at least in Saunders Is there is a lot of carbonate rocks. It could substantially impact radiocarbon dating ('old carbon effect), since bulk sediment/peat samples were dated. How did the authors did overcome this issue?

REPLY: Indeed, according to Kurtz (1950) Saunders Island is mainly composed of siltstones, shales and dolomite of the Narssârssuk formation, which is described by Dawes (2006) to be composed of pale carbonates and siliciclastic redbeds of the Narssârssuk Group. Although carbonatic rock is present in the study area we leave out of account its potential impact on the selected peat mosses, which were radiocarbon-dated in the present study, as described in detail by Davidson et al. (2018). We did not date bulk sediment samples.

What sediment/rocks were in the cores' basement? Are peat profile represent entire

biogenic sedimentation on the spots?

REPLY: This concern was also raised by ref#2. We stopped drilling when we hit boulders, larger than the drill tube diameter, whose density we assume to increase at the transition between the overlying peat and the underlying bedrock. We therefore assume the lowermost (oldest) peat to be close to the bedrock surface. Thus, we almost captured the entire peat profile at the given location, although we did not probe the bedrock basement. We added the following specification to section 3.1: "Extensions were used to reach deeper deposits until the corer hit boulders, larger than the drill tube diameter, whose density increased at the transition between the peat and the underlying bedrock."

2. How were the cores divided into zones/periods? What were the criteria? Was the division evaluated with statistical analysis (is it significant?)

REPLY: We added to section 3.3 the following information: "The zonation of the cores was deduced from obvious changes in the accumulation rates."

3. Please give more detail on subfossil Testacea analysis methodology, namely how big samples were taken to analysis, counted quantities etc. I would see 'classical' percentage diagrams rather.

REPLY: We added the following information to section 3.4: "Samples of about 1 g (dry weight) for testacean analysis were suspended in purified water and wet-sieved through a 500-$\mu$m screen." The general agreement is that direct counting recovers from a sample only a certain fraction of species diversity. Thus, higher numbers of identified specimens per sample enlarge the probability that all species present in the assemblage are captured. However, in testacean research, this approach entails an enormous effort for samples with low shell density. The standard quantity of observed individuals customary in protistology is 160 specimens, but this amount is not applicable to all fossil communities. The quantity, i.e. the number of observed testacean individuals, depends on the original community species richness, on the presence of

dominant species and species abundance structure, on the density of testacean organisms in the soil, and on the amount of shells damaged owing to fossilisation processes. Also, patchy distribution of testate amoebae reflecting habitat heterogeneity may affect the process of recovering species. In fossil samples with poor testacean density, the interpretation focuses on the ecological groups combining species with similar requirements in habitat and environmental conditions. Due to low shell densities in the studied cores, we chose this approach, leading to the presentation of ecological groups in Figures 4 and 5 and cautious interpretation of these records in section 5.1. A percentage diagram seems useless due to the low count numbers per sample. We added the count number as Supplementary Material (Tables S1 and S2) and the following information to section 3.4: "Count numbers of testacean shells per sample were generally low (Tables S1 and S2). Thus, the interpretation is cautiously based on the ecological groups combining species with similar requirements under specific habitat and environmental conditions."

4. The values presented in the Table 2 (results for food and faeces) cannot have uncertainty 0.0. Please report at least measurement uncertainties. I hope, that in means calculation authors include measurement uncertainty and applied 'error propagation rule' for mean uncertainty calculations.

REPLY: We deleted the standard deviation of ±0.0 for those samples where only one measurement was undertaken (n = 1) from Table 2. The analytical accuracies of the respective methods are given in section 3.3 where we added the following information: "The OM analyses of the modern source material (bird's food and faeces samples, Table 2) were undertaken at the UC Davis Stable Isotope Facilities, California, USA, whose lab procedures are described in detail at https://stableisotopefacility.ucdavis.edu/. The long-term standard deviation is 0.2 ‰ for $\delta$13C and 0.3 ‰ for $\delta$15N." For the modern reference and core data (per core zone and for the entire core) we applied once the calculation of mean values and standard deviations as presented in Table 2 to capture the data range for comparison. Propagation of uncertainty is un-applicable because the mean values were not included in further calculations.

Table 2: in row all GL zones should be '4400 to 540'

REPLY: Changed accordingly.

Page 1, row11: 'ground at sea' sounds tricky, consider change to 'area'

REPLY: Changed accordingly.

Page 5, row2: please report accuracy in ‰

REPLY: Changed accordingly.

References to our replies Davidson, T.A., Wetterich, S., Johansen, K.L., Grønnow, B., Windirsch, T., Jeppesen, E., Syvyränta, J., Olsen, J., González-Bergonzoni, I., Strunk, A., Larsen, N.K., Meyer, H., Søndergard, J., Dietz, R., Eulears, I., Mosbech, A.: The history of seabird colonies and the North Water ecosystem: Contributions from palaeoecological and archaeological evidence, Ambio, 47 (Suppl 2), 175-192, 2018. Dawes, P.R.: Explanatory notes to the Geological map of Greenland, 1:500000, Thule, Sheet 5, Geological Survey of Denmark and Greenland Map Series 2, 97 pp. + map. GEUS Copenhagen, Denmark, 2006. Kurtz, V.E.: Geology of the Thule area, Greenland. Proceedings of the Oklahoma Academy of Science, 31, 83-89, 1950.

**Table S1** Testacean specimen counts per sample from the Annikitsoq core (GL3), including ecological indication of the different taxa.

| core GL3 | eurybiontic | | | | | | | | | moss-xerophilic | | | | | | |
|---|---|---|---|---|---|---|---|---|---|---|---|---|---|---|---|---|
| Depth interval [cm] | Centropyxis aerophila | Centropyxis constricta | Centropyxis constricta v. minima | Schoenbornia humicola | Cyclopyxis eurystoma | E. laevis | Trinema complanatum | Trinema lineare | Trinema lineare v. terricola | Arcella arenaria v. compressa | Assulina muscorum | Corythion dubium | Corythion dubium v. minima | Corythion dubium v. orbicularis | Corythion pulchellum | Valkanovia elegans |
| 0-5 | | | | | | 1 | | | | | 40 | | 2 | | | |
| 5-10 | | | | | | | | | | | 4 | | | | | |
| 10-15 | | | | | | | | | | | | | 2 | | | |
| 15-20 | | | | | | | | | | | 14 | 1 | | | | |
| 20-24 | | | | | | | | | | | 8 | 1 | | | | |
| 28-31 | | | | 1 | | | | | | | 5 | | | | | |
| 36-40 | | | | | | | | | | | 3 | | | | | |
| 48-52 | | | | | | | | | | | | | | | | |
| 60-65 | | | | | | | | | | | | | | | | |
| 69-73 | | | | | | | | | | | | | | | | |
| 77-81 | | | | | | | | | | | | | | | | |
| 87-91 | | | | 1 | | | | | | | | | | | | |
| 99-102 | | | | | | | | | | | | 1 | | | | 1 |
| 110-115 | | | | | | | | | | | | | | | | |
| 119-123 | | | | | | 1 | | | | | | | | | | |
| 127-131 | | | | | | | | | | | | 1 | | | | |
| 135-138 | | | | | | | | | | | | 1 | | | | |
| 142-145 | | | | | | | | | | | | 2 | | 1 | | 1 |
| 149-153 | | | | | | | | | | | | | | | | |
| 157-164 | | | | | | | | | | | | | | | | |
| 168-172 | | | | | | | | | | | | | | | | |
| 180-184 | | | | | | | | | | | | | | | | |
| 191-198 | | | | | | | | | | | | | | | | |
| 201-205 | | | | | | | | | | | | | | | 1 | |
| 212-217 | | | | | 1 | | | | | 1 | | 3 | | | 1 | 2 |
| 221-224 | | 1 | | | | | 2 | | | 2 | 5 | | | | | |
| 227-231 | | | | | | | | | | | 2 | 1 | | | 3 | 4 |
| 235-242 | | | | | | | | 3 | | | | | | 1 | 1 | 3 |
| 248-252 | | | | | | | | 2 | | | | | | | | |
| 260-265 | | | | | | 1 | | 1 | | 1 | 1 | | 1 | | 1 | 3 |
| 257-271 | | | | | | | | | | | | 2 | | | | 7 |
| 280-285 | | | | | | | | | | | | 2 | | | | 8 |
| 290-295 | | | | | | | | | | | 1 | 3 | | | | 14 |
| 300-305 | | | | | | | | 5 | 1 | | | 10 | | 10 | | |
| 309-313 | | | | | | | | 8 | | | 1 | | 2 | 1 | | 3 |
| 316-320 | 1 | | 1 | | | | 1 | 3 | | | | 3 | | | | 2 |

| core GL3 | moss-hygrophilic | | | | | | | | | | | | | | | | | | | | hydrobiontic | | | | |
|---|---|---|---|---|---|---|---|---|---|---|---|---|---|---|---|---|---|---|---|---|---|---|---|---|---|
| Depth interval [cm] | Centropyxis elongata | Euglypha aspera | Euglypha compressa f. glabra | Euglypha cf. denticulata | Euglypha strigosa | Euglypha strigosa f. glabra | Nebela collaris | Nebela lageniformis | Nebela parvula | Nebela tincta | Pseudodifflugia gracilis | Pseudodifflugia gracilis v. terricola | Pseudodifflugia fascicularis | Pseudodifflugia cf. horrida | Pseudodifflugia fulva | Pseudodifflugia jungi | Pseudodifflugia sp. 1 | Pseudodifflugia sp. 2 | Trinema complanatum v. elongata | Trinema complanatum v. platystoma | Difflugia bacillariarum | Difflugia lucida | Difflugia pristis | **number of specimens** | **number of taxa** |
| 0-5 | | | | 1 | | | | | | | 4 | 3 | | | 1 | 1 | | | | | | | | 52 | 8 |
| 5-10 | | | | 1 | | | | | | | 13 | 2 | | | 4 | 1 | 1 | | | | | | | 26 | 7 |
| 10-15 | | | | 1 | | | | | | | 17 | | | | 10 | 1 | | | | | | | | 31 | 5 |
| 15-20 | | | | 1 | | | | | | | 15 | 4 | | | 3 | | 3 | | | | | | | 41 | 7 |
| 20-24 | | | 1 | | | | | | | | 14 | | | | 2 | 1 | | | | | | | | 27 | 6 |
| 28-31 | 1 | | | | | | | | | | 13 | | | | 2 | 2 | | | | | | | | 23 | 6 |
| 36-40 | | | 1 | | | 1 | 3 | | 1 | | 13 | 1 | | 1 | 12 | 2 | | 1 | | | | | | 39 | 11 |
| 48-52 | | | | | | 2 | 3 | | | 6 | 17 | 1 | | | 7 | | | 1 | | | | | | 37 | 7 |
| 60-65 | | | | | | | 3 | | 2 | 2 | 14 | 1 | | | 5 | | | 1 | | | | | | 28 | 7 |
| 69-73 | | | | | | | | | | 2 | 10 | | | | 1 | | | 1 | | | | | | 14 | 4 |
| 77-81 | | | | | | 1 | | | | | 10 | | | | 0 | | | | | | | | | 11 | 2 |
| 87-91 | | | 1 | | | 1 | 1 | | 2 | 1 | 10 | 4 | 1 | | 1 | | | 2 | | | | | | 24 | 11 |
| 99-102 | | | | | | | 1 | | | | 1 | 1 | | | | | | | | | | | | 5 | 5 |
| 110-115 | | | 1 | | | 1 | | | 2 | 4 | 12 | | | | | | | | | | | | | 20 | 5 |
| 119-123 | | | | | | 1 | 9 | | 3 | 8 | 16 | 2 | | | 3 | | | | | | | | | 42 | 8 |
| 127-131 | | | | | | | | | | 4 | 1 | | | | | | | | | | | | | 6 | 3 |
| 135-138 | | | | | | | | | | 26 | 15 | | | | | | | | | | | | | 42 | 3 |
| 142-145 | | | | | | | 2 | | | 12 | 24 | 1 | | | | | | | | | | | | 43 | 7 |
| 149-153 | | | | | | | | | | 2 | 5 | | | | | | | | | | | | | 7 | 2 |
| 157-164 | | 1 | 1 | | | 1 | | | | | 9 | | | | 3 | 1 | | 1 | | | 3 | 2 | | 22 | 9 |
| 168-172 | | | | | | | | | | 1 | 35 | | | | 11 | | | 1 | | | 1 | | 1 | 50 | 6 |
| 180-184 | | 2 | 2 | | | 5 | 1 | | 2 | 1 | 21 | | | | 7 | | | 1 | | | | | | 42 | 9 |
| 191-198 | | 1 | 1 | | 3 | | | | | 5 | 19 | | | | 8 | | | 0 | | | | | | 37 | 6 |
| 201-205 | | 2 | 2 | | | 6 | | 1 | | 8 | 25 | | | | 15 | | | 0 | | | | | | 60 | 8 |
| 212-217 | | 2 | 3 | 1 | | 12 | | | 1 | 2 | 41 | | | | | | | 1 | 1 | | | | | 71 | 14 |
| 221-224 | | | 2 | 1 | | 10 | | | | | 17 | | | | 10 | | | 1 | 4 | 1 | | | | 53 | 12 |
| 227-231 | | 1 | 1 | | | 2 | | | | | 9 | | | | 4 | 1 | | | | | | | 1 | 29 | 11 |
| 235-242 | | | | 1 | | 11 | | | | | 12 | 1 | | | 2 | | | 1 | | | | | | 33 | 10 |
| 248-252 | | | 3 | | | 7 | | | | | 5 | | | | 2 | | | | | | | | | 17 | 5 |
| 260-265 | | | 2 | 2 | | 13 | | | | | 15 | 1 | | | 6 | | | | | | | | | 46 | 13 |
| 257-271 | | | 2 | | | 3 | | | | | 3 | | | | | | | | | | | | | 17 | 5 |
| 280-285 | | | 1 | 1 | | 5 | | | | 1 | | | | | | | | | | | | | | 18 | 6 |
| 290-295 | | 1 | 3 | | | 1 | | | | | 1 | | | | | | | | | | | | | 24 | 7 |
| 300-305 | | | 1 | | | 1 | | | | | | | | | | | | | | | | | | 22 | 6 |
| 309-313 | | | | | | | | | | | | | | | | | | | | | | | | 7 | 5 |
| 316-320 | | 1 | 4 | | | 8 | | | | | | | | | | | | | | | | | | 18 | 9 |

**Fig. 1.** Table S1

**Table S2** Table S2 Testacean specimen counts per sample from the Appat core (SI1), including an ecological indication of the different taxa.

| core SI1 | Eurybiontic | | | | | | | | | | moss-xerophilic | moss-hygrophilic | hydrobiontic | | |
|---|---|---|---|---|---|---|---|---|---|---|---|---|---|---|---|
| Depth inverval [cm bs] | Centropyxis aerophila | Centropyxis aerophila v. minuta | Centropyxis constricta | Centropyxis constricta v. minima | Centropyxis sylvatica | Centropyxis sylvatica v. minor | Cyclopyxis eurystoma | Cyclopyxis eurystoma v. parvula | Plagiopyxis cf. callida | Trinema lineare | Arcella arenaria v. compressa | Nebela tincta | Arcella discoides var. difficilis | **number of specimens** | **number of taxa** |
| 0-10 | **1** | **1** | | **10** | | **4** | | **6** | | | **2** | | | **24** | **6** |
| 10-13 | **1** | | **4** | | **6** | **2** | | **1** | | | **3** | | | **17** | **6** |
| 19-22 | | | **1** | | **1** | **1** | **1** | | | | | | | **4** | **4** |
| 26-30 | | | | | **7** | | | | | **1** | | | | **8** | **2** |
| 34-38 | | | **1** | | **1** | | **1** | **1** | | | | | | **4** | **4** |
| 42-46 | | | **1** | | | | | | | | | | | **1** | **1** |
| 52-56 | | | | | **1** | | | **1** | | | **1** | | | **3** | **3** |
| 64-68 | | | | | **1** | | **1** | **1** | | | | | | **3** | **3** |
| 72-76 | | | **1** | | **1** | | | | | | | | | **2** | **2** |
| 80-84 | | | | | **1** | | **3** | | **1** | | | | | **5** | **5** |
| 88-92 | | | | | **1** | | | | | | | | | **1** | **1** |
| 96-100 | | | **1** | | **1** | | | | | | **1** | | | **3** | **3** |
| 103-107 | | | **1** | | **3** | | **1** | | | | | | | **5** | **3** |
| 115-120 | | | **3** | | **3** | | | | | | **1** | | | **7** | **3** |
| 128-132 | | | | | **1** | | | | | | | | | **1** | **1** |
| 136-140 | | | **5** | | **1** | | | | | | | | | **6** | **2** |
| 144-148 | **1** | | **3** | **1** | **1** | | **3** | | | | **3** | | | **12** | **6** |
| 150-154 | **1** | | **1** | | | | | | | | **1** | | | **3** | **3** |
| 168-172 | | | | | | **1** | | | | | | | | **1** | **1** |
| 176-180 | | | **1** | | | | | | | | **1** | | | **2** | **2** |
| 195-198 | | | | | | | | | | | | 1 | 1 | **2** | **2** |

**Fig. 2.** Table S2

---

## Author Comment (AC2) · 12 Sep 2019

Comments on manuscript by Wetterich et al. submitted to Biogeosciences. The manuscript is well written and fairly well illustrated and the subject is suitable for Biogeosciences.

REPLY: We are grateful to referee #2 for the time and effort spent on reviewing our manuscript.

However, the authors need to refer to previous work carried out in the area. Over 30 radiocarbon ages from peat deposits below sea-bird colonies from the Thule region

[Figure]

were reported by Delibrias et al. (1972) and discussed by Malaurie et al. (1972). The peat deposits were also discussed by Salomonsen (1979). The authors could also refer to a paper by Burnham et al. (2009). Malaurie et al. discussed peat accumulation rates – an issue also discussed by Wetterich et al. The oldest ages reported by Malaurie were ca. 1500 years BP. Wetterich et al. have ages going back to ca. 5000 years. I wonder if the lack of older ages is due to the sampling methods? I would assume that sea birds colonised NW Greenland already in the Early Holocene, but I agree that colonies expanded during the Late Holocene. We know from radiocarbon dating of shells of marine molluscs in the region that there was a rich marine invertebrate fauna already in the Early Holocene. And of course there are also very large sea-bird colonies in other parts of Greenland.

REPLY: We agree that Early Holocene seabird colonisation in NW Greenland is likely for some species although dated records are very rare. Arrival dates as old as in our study, with about 5.6 cal kyr BP for thick-billed murre at Appat (Saunders Island) and 4.4 cal kyr BP for little auk at Annikitisoq (north-west of Cape York), have not been reported previously for these species in the Baffin Bay region. Peat deposits on Nord-vestø of the Carey Islands (also in the North Water polynya (NOW) region, likely related to Atlantic Puffin (Fratercula arctica) colonies, were dated to 7.1 to 5.1 cal kyr BP (Ben-nike et al., 2008). For this record, Outridge et al. (2016) emphasise the role of marine derived nutrients (MDN) and trace element input from seabirds for peat formation over 2000 years. A similar peat formation from Nordvestø was dated between 6.3 and 4.4 kyr BP, but it was mainly studied with focus on fossil diatom ecology and the possible relation to seabirds was not considered in the paper (Brown et al., 1994). Burnham et al. (2009) provide evidence of Gyrfalcon (Falco rusticolus) presence in the Thule District from at least 690-530 cal yr BP. The study by Salomonsen (1979) on seabird colony ecology was undertaken in SW Greenland in the Low Arctic Zone (as opposed to our setting in the High Arctic) and lacks therefore also the control of the North Water polynya dynamics on seabird populations that we emphasise in our study. The Nord-vestø 6.3 to 4.4 kyr BP peat record mentioned above is, however, shortly discussed in

Salomonsen (1979). Thus, considering the recommended literature, a direct comparison of our study to Bennike et al. (2008), Outridge et al. (2016), Salomonsen (1979) and Burnham et al. (2009) is not applicable, because other bird species and/or areas and/or time periods were studied. Malaurie et al. (1972) published peat formations from the Thule District related to the same bird species as in our study, but as stated by the authors they most likely did not cover the entire peat sequence at the different sites: "For strictly operational reasons, the depth of the [peat] monoliths did not exceed 82 cms. It is possible, though not certain that [. . .] the sampling pit could have been deepened further and that the earliest phase of peat formation could have been older [. . .]" (pp. 108, Malaurie et al., 1972). We have visited several of the sites sampled by Malaurie et al. (1972), and at least at Savissivik the peat layers are certainly thicker than 82 cm. Thus, the oldest dates with possible seabird control on peat accumulation from Malaurie et al. (1972)/Delibrias et al. (1972) are much younger than in our records, ranging from 1860 yr BP to 1060 yr BP. The peat accumulation rates reported by Malaurie et al. (1972) are relevant for comparison with our findings, but the results treated with caution. In detail, the little-auk-affected peat record of the site Idglolorssuit in Malaurie et al. (1972) suffers from distinct age reversals except for the uppermost and the lowermost samples and seems therefore less useful to deduce accumulation rates. The same is true for Malaurie's peat record from Savigssivik that lacks chronology. Own observations and our interpretation of the site Ivsugissok (Parker Snow Bay) in Malaurie et al. (1972) doubt its alignment to the thick-billed murre colony as stated by the authors. The thick-billed murre colony is situated at a vertical cliff that drops directly into the ocean. Please look at the map inserted below (Figure R1), where the murre colonies are indicated by blue dots. About the location of their peat site, Malaurie et al. (1972) write: "At Ivssugissok (near Cape Dudley Digges) from the first slopes north of the bay 1 km from the glacier . . .". As we understand this description, they would have taken their peat sample close to the actual place name "Issuvissooq", which is just the new spelling of the place name Ivssugissok that they use, or alternatively just north of Cape Dudley Digges. Either way, this is far from the thick-billed murre colony,

and based on our GPS-tracking of murres from this colony no murres would fly over the sites (they head directly from the colony SW to open sea and back). On the other hand, it looks like Malaurie's site could well be from inside or near a little auk colony (the red polygons on the map). In conclusion, we would re-interpret Malaurie's peat deposits at Ivssugissok as being generated by little auks, instead of thick-billed murres. There is a further complication though: Ivssugissok is an old abandoned Inuit settlement. If the peat samples were extracted close to the settlement, peat growth might also have been influenced by nutrients brought to the site by humans, at least from 1200/1400 AD and onwards. From Malaurie's publication, it appears that they extracted the samples at Ivssugissok in April. The birds do not arrive in the colonies till early/mid May, so that may be the reason why they have misinterpreted the situation. For these reasons, only the little auk colony site at Iita from Malaurie's dataset (which is also the longest record) seems appropriate to refer to, which we did as follows in section 5.1: "The spatial distribution of the polygonal peat development is related to seabird colonies. Peat records published by Malaurie et al. (1972) from a little auk colony site at Iita at the northern edge of the NOW polynya indicate continuous peat growth since 1795 cal yr BP years (Delibrias et al., 1972; calibrated using INTCAL, CALIB REV7.1.0, Stuiver and Reimer, 1993; Reimer et al., 2013) at a mean accumulation rate of 37 yr cm-1 over the entire section of 52 cm. This rate is similar to the value of 38 $\pm$ 12 yr cm-1 obtained from the uppermost GL3-III core zone of the Annikitisoq core dated from 2620 to 540 cal yr BP (Table 2). Malaurie et al. (1972) interpreted their data as "... the rate of growth of the peat deposits shows high degree of dependence upon local conditions ..." This is supported by the present study, which also emphasises variation in peat accumulation over time dependent on local conditions."

The authors describe some of the cores as brown moss peat. I wonder if the mosses were identified? The peat on Nordvestø consists of Aplodon wormskioldii, which is a most uncommon fossil species. I wonder if the other peat deposits in the Thule region are also dominated by this species? On page 10, line 35 we hear about Polytrichastrum alpinum, which is present in the GL3 core. Does it dominate?

REPLY: We identified only exemplarily some on the mosses for dating issues. In this context, Polytrichastrum alpinum was observed in core GL3, but it remains unclear whether it dominates or not.

Page 2, line 8. How many pairs of thick-billed murre?

REPLY: Added to section 1: "The thick-billed murre population of the Thule District is estimated to approx. 225.000 breeding pairs, accounting for 68 % of the Greenlandic breeding population of the species (Merkel et al. 2014)."

Page 2, line 33 and 34. I prefer use rather than employ

REPLY: Changed accordingly.

Page 3, line 18. Please provide data on the elevation of the sites. Are the sites below or above the marine limit?

REPLY: Added to section 2: "Based on the GIMP Digital Elevation Model (Howat et al., 2014) and the geoid EGM2008, sample sites GL3 and GL3-IW at Annikitisoq are 22.1 m above mean sea level, whereas sample site SI1 on Appat is 14.2 above mean sea level. All three sites are well above the marine limit."

Page3, line 24. Please provide data on the local vegetation as you observed in the field. Grasses and sedges are much more important than rushes in the area. Not sure about dwarf shrubs at the actual sampling sites.

REPLY: Unfortunately, we did not conduct a systematic description of the current vegetation at the sample sites. We added our field observations to section 2 as follows: "In the little auk colony at Annikitsoq, the general vegetation pattern corresponded very closely to the vegetation pattern in the little auk colony of Qeqertaq (Salve Island), which is described in detail in Mosbech et al. (2018) and seems to be representative of most little auk colonies in Southern Thule without muskoxen. Thus, up high, close to the boulder scree where the little auks nest, the vegetation was dominated by foxtail grass and chickweed, whereas further down the slope, where the GL3 core

was extracted, the vegetation solely consisted of mosses and lichens (corresponding approx. to the 150 m mark on the vegetation transect in Fig. 3 of Mosbech et al., 2018). Mosses dominated the centre of the GL3 polygon, where the coring took place, whereas the lichens dominated the surface at the periphery of the polygon (Fig. 2a-c). We observed no dwarf shrubs at Annikitsoq, and it is our general impression that dwarf shrubs are absent in areas under heavy influence of little auks. The vegetation at the sampling site at Appat was more diverse, with mosses and foxtail grass at the centre of the polygon, where the SI1 core was extracted, and a mixed community at the periphery of the polygon consisting of Arctic willow, foxtail grass, chickweed and Arctic poppy (Fig. 2e)."

Page 4, line 2. How do you know you reached bedrock? Could it be stones or boulders? From the photos it appears that both sampling sites are located in areas with thick gravel deposits.

REPLY: This concern was also raised by ref#1. We stopped drilling when we hit boulders, larger than the drill tube diameter, whose density we assume to increase at the transition between the overlying peat and the underlying bedrock. We therefore assume the lowermost (oldest) peat to be close to the bedrock surface. Thus, we almost captured the entire peat profile at the given location, although we did not probe the bedrock basement. We added the following specification to section 3.1: "Extensions were used to reach deeper deposits until the corer hit boulders, larger than the drill tube diameter, whose density increased at the transition between the peat and the underlying bedrock."

Page 5, line 4-10. More information on testacean analysis is needed. How many specimens were counted per sample? What was the sample size? It would be nice to see some testacean diagrams.

REPLY: We added the following information to section 3.4: "Samples of about 1 g (dry weight) for testacean analysis were suspended in purified water and wet-sieved

through a 500-$\mu$m screen." The general agreement is that direct counting recovers from a sample only a certain fraction of species diversity. Thus, higher numbers of identified specimens per sample enlarge the probability that all species present in the assemblage are captured. However, in testacean research, this approach entails enormous effort for samples with low shell density. The standard quantity of observed individuals customary in protistology is 160 specimens, but this amount is not applicable to all fossil communities. The quantity, i.e. number of observed testacean individuals, depends on the original community species richness, on the presence of dominant species and species abundance structure, on the density of testacean organisms in soil, and on the amount of shells damaged owing to fossilisation processes. Also patchy distribution of testate amoebae reflecting habitat heterogeneity may affect the process of recovering species. In fossil samples with poor testacean density, the interpretation focuses on the ecological groups combining species with similar requirements in habitat and environmental conditions. Due to low shell densities in the studied cores, we chose this approach leading to the presentation of ecological groups in Figures 4 and 5 and cautious interpretation of these records in section 5.1. A percentage diagram seems useless due to the low count number per sample. We added the count number as Supplementary Material (Tables S1 and S2) and the following information to section 3.4: "Count numbers of testacean shells per sample were generally low (Tables S1 and S2). Thus, the interpretation is cautiously based on the ecological groups combining species with similar requirements under specific habitat and environmental conditions."

Page 5, line 14. The succession was divided into three zones

REPLY: Changed accordingly.

Page 5, line 28. Is there only one testacean assemblage in the whole core?

REPLY: Changed to "testacean record" while the differing species assemblages along the core are described further down the ms text.

Page 7, line 3-4. Intense deglaciation occurred in the Early Holocene, long before the
Holocene Thermal Maximum – due to abrupt warming at the Younger Dryas-Holocene transition. With respect to the deglaciation chronology see Bennike & Björck (2002).

REPLY: Changed accordingly to: "The abrupt warming at the Younger Dryas-Holocene transition promoted intense deglaciation in Arctic Canada and Greenland (Briner et al., 2016) and left large parts of the northern Baffin Bay coastlines free of glacial ice from the Early Holocene (Bennicke and Björck, 2002)."

Page 7, line 5 9.9 and 9.2 kyr BP – not yr BP!

REPLY: Changed accordingly.

Page 7, line 3-8. According to the compilation by Bennike & Björck (2002) the two study sites were deglaciated before 10 cal. kyr BP

REPLY: We appreciate the literature recommendation of Bennike & Björck (2002). Indeed, Bennike & Björck (2002) present radiocarbon dates of Early Holocene deglaciation from the Wolstenholme Fjord area of 10.6 and 11.2 cal kyr BP. We therefore added the following statement to section 5.1: "Early Holocene deglaciation of the Wolstenholme Fjord area was dated to 11.2 and 10.6 cal kyr BP (Bennike and Björck, 2002). Radiocarbon-dated marine bivalves in raised marine deposits on Appat although at a different location as our study site show ages between about 9.9 and 9.2 cal kyr BP (Farnsworth et al., 2018) in agreement with modelled deglaciation data for the Greenland ice sheet according to which both Appat and Annikitisoq became ice free around 10 to 9 cal kyr BP (Lecavalier et al., 2014).".

Page 11, line 23. I think the main difference is the number of little auks compared with thick-billed murre. Also the different nesting places as mentioned by the authors.

REPLY: Agree. We added the following statement to section 5.2 to capture this point: "The difference in peat accumulation rates between the sites might relate to the different colony sizes, the little auk colony being much larger than the thick-billed murre colony, as well as to the different preferences of the two bird species with regard to

location of breeding sites."

References to our replies Bennike, O., Björck, S.: Chronology of the last recession of the Greenland Ice Sheet, Journal of Quaternary Science, 17, 211-217, 2002. Bennike, O., Goodsite, M., Heinemeier, J.: Palaeoecology of Holocene peat deposits from Nordvestø, north-west Greenland, Journal of Paleolimnology, 40, 557-565, 2008, DOI: 10.1007/s10933-007-9181-3 Brown, K.M., Douglas, M.S.V., Smol, J.P.: Siliceous microfossils in a Holocene, High Arctic peat deposit (Nordvestø, northwestern Greenland), Canadian Journal of Botany, 72, 208-216, 1994. Burnham, K.K., Burnham, W.A. & Newton, I.: Gyrfalcon Falco rusticolus post-glacial colonization and extreme long-term use of nest-sites in Greenland, Ibis, 151, 514-522, 2009. Delibrias, G., Guillier, M.T., Labeyrie, J.: Gif natural radiocarbon measurements VII, Radiocarbon, 14, 280-320, 1972. Howat, I.M., Negrete, A., Smith, B.E.: The Greenland Ice Mapping Project (GIMP) land classification and surface elevation datasets, The Cryosphere, 8, 1509-1518, 2014. Malaurie, J., Vasari, Y., Hyvarinen, H., Delibrias, G., Labeyrie, J.: Preliminary remarks on Holocene paleoclimates in the regions of Thule and Inglefield Land, above all since the beginning of our own era, Acta Universitatis Ouluensis, Series A, Scientiae Rerum Naturalium 3, Geologica 1, 105-136, 1972. Merkel, F.R., Labansen, A.L., Boertmann, D.M., Mosbech, A., Egevang, C., Falk, K., Linnebjerg, J.F., Frederiksen, M., Kampp, K.: Declining trends in the majority of Greenland's thick-billed murre (Uria lomvia) colonies 1981-2011. Polar Biology, 37, 1061-1071, 2014. Mosbech, A., Johansen, K.L., Davidson, T.A., Appelt, M., Grønnow, B., Cuyler, C., Lyngs, P., Flora, J.: On the crucial importance of a small bird: The ecosystem services of the little auk (Alle alle) population in Northwest Greenland in a long-term perspective, Ambio, 47(Suppl 2), 226-243, 2018. Outridge, P.M., Goodsite, M.E., Bennike, O., Rausch, N., Shotyk, W.: Seabird transfer of nutrients and trace elements from the north water polynya to land during the mid-holocene warm period, Carey islands, northwest Greenland. Arctic 69(3), 253-265, 2016. Salomonsen, F.: Ornithological and ecological studies in southwest Greenland (59°46'–62°27'N. Lat.), Meddelelser om Grønland, 204(6), 1-214, 1979.

[Figure]

[Figure]

Figure R1: Seabird colonies at Ivsugissooq (Parker Snow Bay).

**Fig. 1.** Figure R1

**Table S1** Testacean specimen counts per sample from the Annikitsoq core (GL3), including ecological indication of the different taxa.

| core GL3 | eurybiontic | | | | | | | | | moss-xerophilic | | | | | | |
|---|---|---|---|---|---|---|---|---|---|---|---|---|---|---|---|---|
| Depth interval [cm] | Centropyxis aerophila | Centropyxis constricta | Centropyxis constricta v. minima | Schoenbornia humicola | Cyclopyxis eurystoma | E. laevis | Trinema complanatum | Trinema lineare | Trinema lineare v. terricola | Arcella arenaria v. compressa | Assulina muscorum | Corythion dubium | Corythion dubium v. minima | Corythion dubium v. orbicularis | Corythion pulchellum | Valkanovia elegans |
| 0-5 | | | | | | 1 | | | | | 40 | | 2 | | | |
| 5-10 | | | | | | | | | | | 4 | | | | | |
| 10-15 | | | | | | | | | | | | | 2 | | | |
| 15-20 | | | | | | | | | | | 14 | 1 | | | | |
| 20-24 | | | | | | | | | | | 8 | 1 | | | | |
| 28-31 | | | | 1 | | | | | | | 5 | | | | | |
| 36-40 | | | | | | | | | | | 3 | | | | | |
| 48-52 | | | | | | | | | | | | | | | | |
| 60-65 | | | | | | | | | | | | | | | | |
| 69-73 | | | | | | | | | | | | | | | | |
| 77-81 | | | | | | | | | | | | | | | | |
| 87-91 | | | | 1 | | | | | | | | | | | | |
| 99-102 | | | | | | | | | | | | 1 | | | | 1 |
| 110-115 | | | | | | | | | | | | | | | | |
| 119-123 | | | | | | 1 | | | | | | | | | | |
| 127-131 | | | | | | | | | | | | 1 | | | | |
| 135-138 | | | | | | | | | | | | 1 | | | | |
| 142-145 | | | | | | | | | | | | 2 | | 1 | | 1 |
| 149-153 | | | | | | | | | | | | | | | | |
| 157-164 | | | | | | | | | | | | | | | | |
| 168-172 | | | | | | | | | | | | | | | | |
| 180-184 | | | | | | | | | | | | | | | | |
| 191-198 | | | | | | | | | | | | | | | | |
| 201-205 | | | | | | | | | | | | | | | 1 | |
| 212-217 | | | | | 1 | | | | | 1 | | 3 | | | 1 | 2 |
| 221-224 | | 1 | | | | | 2 | | | 2 | 5 | | | | | |
| 227-231 | | | | | | | | | | | 2 | 1 | | | 3 | 4 |
| 235-242 | | | | | | | | 3 | | | | | | 1 | 1 | 3 |
| 248-252 | | | | | | | | 2 | | | | | | | | |
| 260-265 | | | | | | 1 | | 1 | | 1 | 1 | | 1 | | 1 | 3 |
| 257-271 | | | | | | | | | | | | 2 | | | | 7 |
| 280-285 | | | | | | | | | | | | 2 | | | | 8 |
| 290-295 | | | | | | | | | | | 1 | 3 | | | | 14 |
| 300-305 | | | | | | | | 5 | 1 | | | 10 | | 10 | | |
| 309-313 | | | | | | | | 8 | | | 1 | | 2 | 1 | | 3 |
| 316-320 | 1 | | 1 | | | | 1 | 3 | | | | 3 | | | | 2 |

| core GL3 | moss-hygrophilic | | | | | | | | | | | | | | | | | | | | hydrobiontic | | | | |
|---|---|---|---|---|---|---|---|---|---|---|---|---|---|---|---|---|---|---|---|---|---|---|---|---|---|
| Depth interval [cm] | Centropyxis elongata | Euglypha aspera | Euglypha compressa f. glabra | Euglypha cf. denticulata | Euglypha strigosa | Euglypha strigosa f. glabra | Nebela collaris | Nebela lageniformis | Nebela parvula | Nebela tincta | Pseudodifflugia gracilis | Pseudodifflugia gracilis v. terricola | Pseudodifflugia fascicularis | Pseudodifflugia cf. horrida | Pseudodifflugia fulva | Pseudodifflugia jungi | Pseudodifflugia sp. 1 | Pseudodifflugia sp. 2 | Trinema complanatum v. elongata | Trinema complanatum v. platystoma | Difflugia bacillariarum | Difflugia lucida | Difflugia pristis | **number of specimens** | **number of taxa** |
| 0-5 | | | | 1 | | | | | | | 4 | 3 | | | 1 | 1 | | | | | | | | 52 | 8 |
| 5-10 | | | | 1 | | | | | | | 13 | 2 | | | 4 | 1 | 1 | | | | | | | 26 | 7 |
| 10-15 | | | | 1 | | | | | | | 17 | | | | 10 | 1 | | | | | | | | 31 | 5 |
| 15-20 | | | | 1 | | | | | | | 15 | 4 | | | 3 | | 3 | | | | | | | 41 | 7 |
| 20-24 | | | 1 | | | | | | | | 14 | | | | 2 | 1 | | | | | | | | 27 | 6 |
| 28-31 | 1 | | | | | | | | | | 13 | | | | 2 | 2 | | | | | | | | 23 | 6 |
| 36-40 | | | 1 | | | 1 | 3 | | 1 | | 13 | 1 | | 1 | 12 | 2 | | 1 | | | | | | 39 | 11 |
| 48-52 | | | | | | 2 | 3 | | | 6 | 17 | 1 | | | 7 | | | 1 | | | | | | 37 | 7 |
| 60-65 | | | | | | | 3 | | 2 | 2 | 14 | 1 | | | 5 | | | 1 | | | | | | 28 | 7 |
| 69-73 | | | | | | | | | | 2 | 10 | | | | 1 | | | 1 | | | | | | 14 | 4 |
| 77-81 | | | | | | 1 | | | | | 10 | | | | 0 | | | | | | | | | 11 | 2 |
| 87-91 | | | 1 | | | 1 | 1 | | 2 | 1 | 10 | 4 | 1 | | 1 | | | 2 | | | | | | 24 | 11 |
| 99-102 | | | | | | | 1 | | | | 1 | 1 | | | | | | | | | | | | 5 | 5 |
| 110-115 | | | 1 | | | 1 | | | 2 | 4 | 12 | | | | | | | | | | | | | 20 | 5 |
| 119-123 | | | | | | 1 | 9 | | 3 | 8 | 16 | 2 | | | 3 | | | | | | | | | 42 | 8 |
| 127-131 | | | | | | | | | | 4 | 1 | | | | | | | | | | | | | 6 | 3 |
| 135-138 | | | | | | | | | | 26 | 15 | | | | | | | | | | | | | 42 | 3 |
| 142-145 | | | | | | | 2 | | | 12 | 24 | 1 | | | | | | | | | | | | 43 | 7 |
| 149-153 | | | | | | | | | | 2 | 5 | | | | | | | | | | | | | 7 | 2 |
| 157-164 | | 1 | 1 | | | 1 | | | | | 9 | | | | 3 | 1 | | 1 | | | 3 | 2 | | 22 | 9 |
| 168-172 | | | | | | | | | | 1 | 35 | | | | 11 | | | 1 | | | 1 | | 1 | 50 | 6 |
| 180-184 | | 2 | 2 | | | 5 | 1 | | 2 | 1 | 21 | | | | 7 | | | 1 | | | | | | 42 | 9 |
| 191-198 | | 1 | 1 | | 3 | | | | | 5 | 19 | | | | 8 | | | 0 | | | | | | 37 | 6 |
| 201-205 | | 2 | 2 | | | 6 | | 1 | | 8 | 25 | | | | 15 | | | 0 | | | | | | 60 | 8 |
| 212-217 | | 2 | 3 | 1 | | 12 | | | 1 | 2 | 41 | | | | | | | 1 | 1 | | | | | 71 | 14 |
| 221-224 | | | 2 | 1 | | 10 | | | | | 17 | | | | 10 | | | 1 | 4 | 1 | | | | 53 | 12 |
| 227-231 | | 1 | 1 | | | 2 | | | | | 9 | | | | 4 | 1 | | | | | | | 1 | 29 | 11 |
| 235-242 | | | | 1 | | 11 | | | | | 12 | 1 | | | 2 | | | 1 | | | | | | 33 | 10 |
| 248-252 | | | 3 | | | 7 | | | | | 5 | | | | 2 | | | | | | | | | 17 | 5 |
| 260-265 | | | 2 | 2 | | 13 | | | | | 15 | 1 | | | 6 | | | | | | | | | 46 | 13 |
| 257-271 | | | 2 | | | 3 | | | | | 3 | | | | | | | | | | | | | 17 | 5 |
| 280-285 | | | 1 | 1 | | 5 | | | | 1 | | | | | | | | | | | | | | 18 | 6 |
| 290-295 | | 1 | 3 | | | 1 | | | | | 1 | | | | | | | | | | | | | 24 | 7 |
| 300-305 | | | 1 | | | 1 | | | | | | | | | | | | | | | | | | 22 | 6 |
| 309-313 | | | | | | | | | | | | | | | | | | | | | | | | 7 | 5 |
| 316-320 | | 1 | 4 | | | 8 | | | | | | | | | | | | | | | | | | 18 | 9 |

**Fig. 2.** Table S1

**Table S2** Table S2 Testacean specimen counts per sample from the Appat core (SI1), including an ecological indication of the different taxa.

| core SI1 | Eurybiontic | | | | | | | | | | moss-xerophilic | moss-hygrophilic | hydrobiontic | | |
|---|---|---|---|---|---|---|---|---|---|---|---|---|---|---|---|
| Depth inverval [cm bs] | Centropyxis aerophila | Centropyxis aerophila v. minuta | Centropyxis constricta | Centropyxis constricta v. minima | Centropyxis sylvatica | Centropyxis sylvatica v. minor | Cyclopyxis eurystoma | Cyclopyxis eurystoma v. parvula | Plagiopyxis cf. callida | Trinema lineare | Arcella arenaria v. compressa | Nebela tincta | Arcella discoides var. difficilis | **number of specimens** | **number of taxa** |
| 0-10 | **1** | **1** | | **10** | | **4** | | **6** | | | **2** | | | **24** | **6** |
| 10-13 | **1** | | **4** | | **6** | **2** | | **1** | | | **3** | | | **17** | **6** |
| 19-22 | | | **1** | | **1** | **1** | **1** | | | | | | | **4** | **4** |
| 26-30 | | | | | **7** | | | | | **1** | | | | **8** | **2** |
| 34-38 | | | **1** | | **1** | | **1** | **1** | | | | | | **4** | **4** |
| 42-46 | | | **1** | | | | | | | | | | | **1** | **1** |
| 52-56 | | | | | **1** | | | **1** | | | **1** | | | **3** | **3** |
| 64-68 | | | | | **1** | | **1** | **1** | | | | | | **3** | **3** |
| 72-76 | | | **1** | | **1** | | | | | | | | | **2** | **2** |
| 80-84 | | | | | **1** | | **3** | | **1** | | | | | **5** | **5** |
| 88-92 | | | | | **1** | | | | | | | | | **1** | **1** |
| 96-100 | | | **1** | | **1** | | | | | | **1** | | | **3** | **3** |
| 103-107 | | | **1** | | **3** | | **1** | | | | | | | **5** | **3** |
| 115-120 | | | **3** | | **3** | | | | | | **1** | | | **7** | **3** |
| 128-132 | | | | | **1** | | | | | | | | | **1** | **1** |
| 136-140 | | | **5** | | **1** | | | | | | | | | **6** | **2** |
| 144-148 | **1** | | **3** | **1** | **1** | | **3** | | | | **3** | | | **12** | **6** |
| 150-154 | **1** | | **1** | | | | | | | | **1** | | | **3** | **3** |
| 168-172 | | | | | | **1** | | | | | | | | **1** | **1** |
| 176-180 | | | **1** | | | | | | | | **1** | | | **2** | **2** |
| 195-198 | | | | | | | | | | | | **1** | **1** | **2** | **2** |

**Fig. 3.** Table S2